# Rapid evolution and host immunity drive the rise and fall of carbapenem resistance during an acute *Pseudomonas aeruginosa* infection

Rachel Wheatley [1,11], Julio Diaz Caballero[1,11], Natalia Kapel[1], Fien H. R. de Winter [2], Pramod Jangir[1], Angus Quinn [1], Ester del Barrio-Tofiño[3], Carla López-Causapé[3], Jessica Hedge [1], Gabriel Torrens[3], Thomas Van der Schalk [2], Basil Britto Xavier [2], Felipe Fernández-Cuenca[4], Angel Arenzana[4], Claudia Recanatini [5], Leen Timbermont [2], Frangiscos Sifakis[6], Alexey Ruzin[7], Omar Ali[7,10], Christine Lammens[2], Herman Goossens[2], Jan Kluytmans[5,8], Samir Kumar-Singh[2,9], Antonio Oliver[3], Surbhi Malhotra-Kumar [2] & Craig MacLean [1✉]

It is well established that antibiotic treatment selects for resistance, but the dynamics of this process during infections are poorly understood. Here we map the responses of *Pseudomonas aeruginosa* to treatment in high definition during a lung infection of a single ICU patient. Host immunity and antibiotic therapy with meropenem suppressed *P. aeruginosa*, but a second wave of infection emerged due to the growth of *oprD* and *wbpM* meropenem resistant mutants that evolved in situ. Selection then led to a loss of resistance by decreasing the prevalence of low fitness *oprD* mutants, increasing the frequency of high fitness mutants lacking the MexAB-OprM efflux pump, and decreasing the copy number of a multidrug resistance plasmid. Ultimately, host immunity suppressed *wbpM* mutants with high meropenem resistance and fitness. Our study highlights how natural selection and host immunity interact to drive both the rapid rise, and fall, of resistance during infection.

[1] University of Oxford, Department of Zoology, Oxford, UK. [2] Laboratory of Medical Microbiology, Vaccine and Infectious Disease Institute, University of Antwerp, Wilrijk, Belgium. [3] Hospital Universitario Son Espases, Palma de Mallorca, Spain. [4] Departamento de Medicina, Universidad de Sevilla, Seville, Spain. [5] Julius Center for Health Sciences and Primary Care, University Medical Center Utrecht, Utrecht University, Utrecht, The Netherlands. [6] Boehringer Ingelheim Pharmaceuticals, Inc, Ridgefield, CT, USA. [7] Microbial Sciences, BioPharmaceuticals R&D, AstraZeneca, Gaithersburg, MD, USA. [8] Microvida Laboratory for Medical Microbiology and Department of Infection Control, Amphia Hospital, Breda, The Netherlands. [9] Molecular Pathology Group, Faculty of Medicine— Laboratory of Cell Biology and Histology, University of Antwerp, Wilrijk, Belgium. [10] Present address: Viela Bio, Gaithersburg, MD, USA. [11] These authors contributed equally: Rachel Wheatley, Julio Diaz Caballero. ✉email: craig.maclean@zoo.ox.ac.uk

Antibiotic resistance has emerged as a serious threat to public health by increasing the health and economic burden associated with bacterial infections[1]. Treating patients with antibiotics selects for resistant bacteria[2,3], and the emergence of resistance during treatment is associated with poorer outcomes in terms of patient health[1,4]. Following treatment, resistance in patients typically returns to baseline levels, although there is considerable heterogeneity in the rate of decline for different microbe/antibiotic combinations[5,6]. Although this link between antibiotic treatment and resistance is straightforward and intuitive, the drivers of evolutionary responses to antibiotic treatment during infections remain poorly characterized. One key challenge in this area is to understand how host immunity impacts resistance. Although it is widely acknowledged that immune responses work in conjunction with antibiotics to suppress bacterial infections, the impact of immunity on evolutionary responses to antibiotics is largely unexplored[7–9].

Progress in understanding the evolution of resistance during infections has largely come from longitudinal sampling of patients suffering from long-term chronic infections associated with diseases such as cystic fibrosis and tuberculosis[10–16]. However, the drivers of resistance in short-term acute infections that cause much of the burden of AMR[17], such as hospital-acquired infections by opportunistic and commensal pathogens, remain poorly understood. Here, we investigate responses to antibiotic therapy through intensive sampling of a single mechanically ventilated patient before, during, and after treatment for a hospital-acquired *Pseudomonas aeruginosa* pneumonia. *P. aeruginosa* is an opportunistic pathogen that is a relatively common cause of nosocomial infections, particularly in immunocompromised patients[18–21], and pneumonia caused by *P.aeruginosa* is associated with a high mortality rate[20]. *P. aeruginosa* infections are difficult to treat with antibiotics due to low outer membrane permeability and the presence of a large repertoire of both intrinsic and acquired resistance mechanisms, including chromosomal mutations and mobile resistance genes[22–24].

To understand the responses to antibiotic treatment, we combined clinical data from the patient with extensive sequencing and phenotypic characterization of isolates that were collected from the lung and gut at regular intervals over a period of 3 weeks. Our clinical data included antibiotic use, bacterial titer data, and host immunity biomarker expression. We collected 12 isolates from each patient sample, and we used whole-genome sequencing and phenotype assays (resistance profiling, fitness) on over 100 isolates to understand the population-level responses to antibiotic therapy and host immunity. Combining these approaches allowed us to understand the population biology of antibiotic resistance during short-term infection at an unprecedented level of resolution.

## Results

**Clinical data.** A 60-year-old patient was admitted to the intensive care unit (ICU) of the Virgen Macarena tertiary care hospital in Seville, Spain with a primary diagnosis of hemorrhagic shock. The patient was intubated and started on mechanical ventilation, and given prophylactic treatment with amoxicillin/acid clavulanic (1000 mg/200 mg IV q8h), which is not effective against *P. aeruginosa*[25,26]. After 72 h of ICU admission, informed consent was obtained and the patient was enrolled in the ASPIRE-ICU study (day 1)[27]. On day 1, the titer of *P. aeruginosa* in the endotracheal aspirate (ETA) was high at $10^6$ colony-forming units per mL (CFU/mL) and *P. aeruginosa* were the only culturable bacteria that were detected in ETA samples (Fig. 1A). A clinical diagnosis of pneumonia was established by the treating physician on day 2 and the patient was treated with piperacillin/tazobactam (4 g/0.5 g

IV q8h for 2 days), meropenem (1 g IV q8h for 2 days) and colistin (3 million IU IV q8h for 13 days) (Fig. 1A). Antibiotic treatment coincided with a dramatic decline in the titer of *P. aeruginosa*, which fell from $>10^4$ CFU/mL at day 2 to $<40$ CFU/mL (assay limit of detection) at day 4. The decline in *Pseudomonas* titer was associated with improved patient health: between day 2 and 7 the sequential organ failure assessment score (SOFA) declined from 14 to 9, and the Clinical Pulmonary Infection Score (CPIS) declined from 8 to 4.

A second wave of *P. aeruginosa* growth was detected between day 8 and day 12, suggesting that the patient suffered either a secondary lung infection or that the extant populations of *P. aeruginosa* recovered. The resurgence of *P. aeruginosa* was accompanied by the establishment of a culturable lung microbiome. This was initially dominated by enteric bacteria (*Enterococcus faecium* and *Klebsiella pneumonia*), but bacteria that are associated with the oral cavity (*Streptococcus orallis*) and skin (*S.epidermidis*) increased in prevalence, eventually replacing *E. faecium* (Fig. 1A). The titer of *Pseudomonas* in this second wave ($10^4-10^5$ CFU/mL) was >10-fold lower than in the initial infection, and no new episodes of clinical pneumonia were reported. Ventilator support was withdrawn on day 23 and no culturable bacteria were detected in ETA samples taken on day 27. The patient was discharged from ICU on day 31 with a SOFA score of 1 and a CPIS of 0.

Intestinal carriage of *P. aeruginosa* was detected upon enrollment, as measured by growth from peri-anal swabs (Fig. 1B). Unlike in the lung, antibiotic treatment (i.e., day 2–4) was not associated with effective suppression of the intestinal population of *P. aeruginosa*. However, the abundance of intestinal *P. aeruginosa* declined rapidly after day 7, and no growth of *P. aeruginosa* was detected in peri-anal swabs that were taken from day 16 onwards (Fig. 1B).

**Phenotypic responses of pulmonary and gut populations to antibiotic treatment.** To gain a better understanding of the role of antibiotics in the dynamics of *P. aeruginosa*, we measured the resistance of lung ($n = 59$) and gut ($n = 48$) isolates to meropenem, piperacillin/tazobactam, and colistin (Fig. 1C–E). Isolates from early time points (day 1–2) had high levels of piperacillin/tazobactam resistance (minimum inhibitory concentration (MIC) > 256 mg/L) that were well above the clinical breakpoint (16 mg/L), suggesting that piperacillin/tazobactam treatment is unlikely to have had any effect on *P. aeruginosa* (Fig. 1B). In contrast, colistin MICs (mean = 0.5 mg/L; s.d = 0; $n = 24$) were below the clinical breakpoints (2 mg/L), suggesting that colistin treatment may have contributed to the suppression of the first wave of lung infection. However, the pulmonary titer of *P. aeruginosa* recovered under continued treatment (days 13–21) without any accompanying increase in colistin resistance (Fig. 1D) or tolerance (Fig. 1E), suggesting that suboptimal pharmacokinetics[28] and/or adaptive changes in gene expression[29], limited the in vivo efficacy of colistin.

Meropenem resistance increased from baseline levels (mean MIC = 9.6 mg/L; s.e.m = 0.677; $n = 24$) following antibiotic treatment (day 13: mean MIC = 29.33 mg/L; s.e.m = 1.04; $n = 12$), suggesting that meropenem treatment suppressed *P. aeruginosa*. However, at the outset of the infection, meropenem resistance was approximately equal to the EUCAST clinical breakpoint concentration (8 mg/L), questioning the efficacy of meropenem. Previous work has shown that synergy exists between colistin and meropenem[30], suggesting that colistin treatment may have increased the efficacy of meropenem. The median meropenem MIC of isolates from the initial infection ($n = 4$ isolates) was reduced by a factor of 4 in the presence of a sub-lethal dose

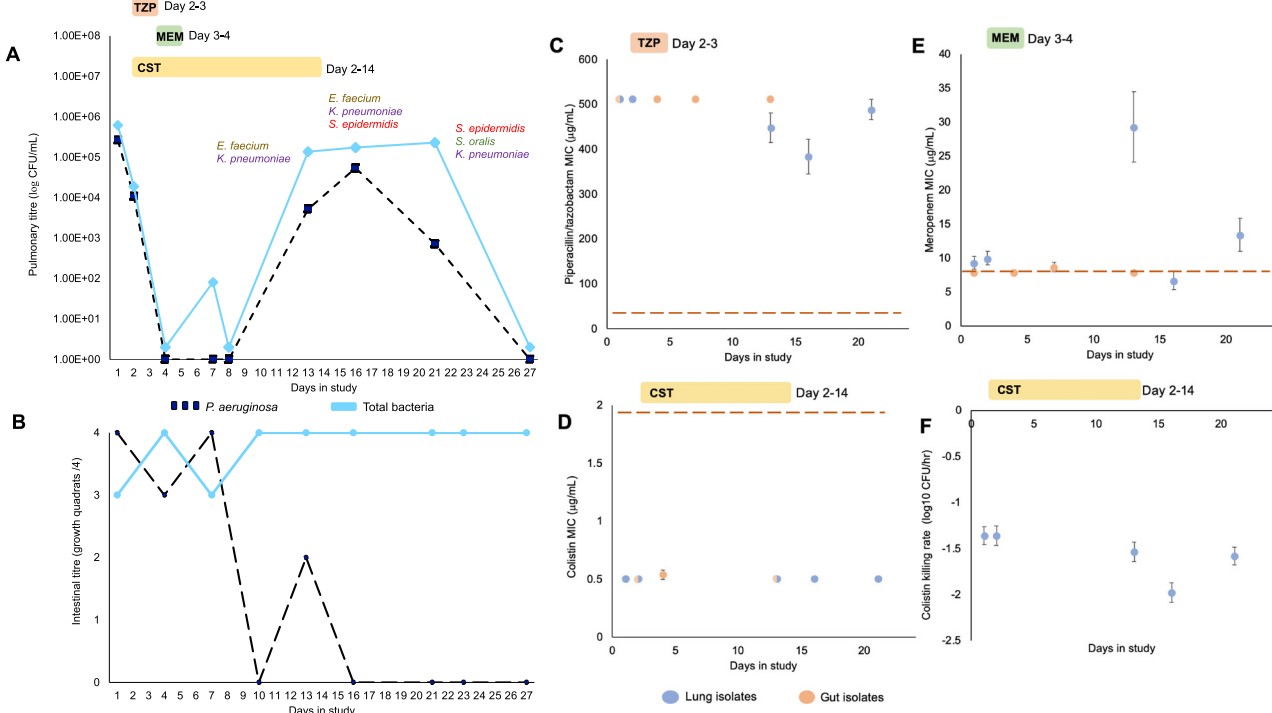

**Fig. 1 Clinical timeline and resistance phenotyping. A** Bacterial abundance in the lung was assessed by plating out samples of endotracheal aspirate (ETA) on *Pseudomonas* selective agar (dark blue) and blood agar (total titer, light blue). Rank-order species abundance data is shown for the total bacterial counts. **B** Bacterial abundance in the gut was assessed on a nominal scale by streaking peri-anal swabs on *Pseudomonas* selective agar and blood agar (total titer). Day 1 in study is 72 h after ICU admission and corresponds to the first day of patient informed consent. The patient received intravenous treatment with piperacillin/tazobactam (TZP: 4 g/0.5 g IV q8h), meropenem (MEM1: g IV q8h) and colistin (CST 3 million IU IV q8h). Panels **C–E** show the mean MIC of lung (blue) and intestinal (beige) isolates, as determined by broth microdilution (+/− s.e.m; $n = 11$ or 12 isolates). Red dashed lines represent the EUCAST clinical breakpoint (01/01/2019 edition). Panel **F** shows the mean rate of change in viable cell titer of lung isolates following treatment with 2 mg/L of colistin (+/− s.e.m; $n = 10$-12 isolates). Source data are provided as a Source Data file.

of colistin (1/2 MIC), suggesting a weak synergistic interaction between these antibiotics. In summary, combining data on bacterial titer during infection with in vitro MIC assays suggests that combination therapy with meropenem and colistin was successful at suppressing the pulmonary population of *P. aeruginosa*, and that the bacterial population responded to treatment by evolving elevated levels of meropenem resistance.

Intestinal isolates that were recovered during or after antibiotic treatment did not have increased resistance to any antibiotics (Fig. 1C–E), providing evidence to support the clinical data showing that antibiotic treatment was not effective against intestinal *Pseudomonas* (Fig. 1B). One potential explanation for this differential effect of antibiotic in the lung and gut is that antibiotic toxicity varies between these anatomical sites. Meropenem diffuses well into lung tissues but is primarily excreted by the renal system as opposed to the biliary system, implying that meropenem concentrations are low in the gut lumen relative to the lung[31]. There is growing evidence that bacterial metabolism plays an important role in determining susceptibility[32], and it is also possible that the efficacy of meropenem differs between the lung and the gut. In this case, the median meropenem $MIC_{50}$ of isolates from the initial infection ($n = 6$) increased from 8 mg/L to 32 mg/L under anaerobic conditions, suggesting that the antibiotic effects of meropenem are contingent on high levels of metabolic activity[32] associated with aerobic metabolism. The pharmacokinetics of colistin in critically ill patients are known to be complex[28,33], making it difficult to make any assumptions about the relative concentration of colistin in the gut compared to the lung, but it is possible that the effective concentration of

colistin in the gut was low relative to the lung. In summary, the available evidence suggests that a combination of low meropenem concentration and high meropenem resistance limited the efficacy of meropenem in the gut.

**Sequencing**. To better understand the rise and fall of resistance, we used a combination of short and long-read sequencing to comprehensively characterize the diversity of *P.aeruginosa*. Initially, we used long-read sequences generated by PacBio to generate a high-quality reference genome for one of the day 1 pulmonary isolates (Fig. 2A). This ST17 reference isolate has a large genome (7,008,585 bp) and a 40 Kb plasmid, p110820, that carries a class 1 integron containing cassettes that confer resistance to aminoglycoside [*aacA4*], ß-lactam [*bla-Oxa10*], and sulfonamide [*sul1*] antibiotics. The reference genome contains mutations that are associated with antibiotic resistance, including target modification mutations that confer resistance to fluoroquinolones (*gyrA* T83I *and parC* S87L) and mutations in repressors of the AmpC ß -lactamase (*ampD* H98Y) and the MexAB-OprM multidrug efflux pump (*nalD*nt58Δ2)[23,34]. We confirmed the overexpression of *ampC* (37.6 ± 21.7-fold) and *mexAB-OprM* (20.8 ± 17.4-fold) in four randomly chosen isolates relative to the PA01 reference strain by quantitative reverse transcription PCR (qRT-PCR). This clone of ST17 has previously been reported as the cause of a nosocomial outbreak at the Virgen Macarena hospital[35].

To identify SNPs, structural variants and copy number variants, we mapped illumina short reads from all 107 isolates

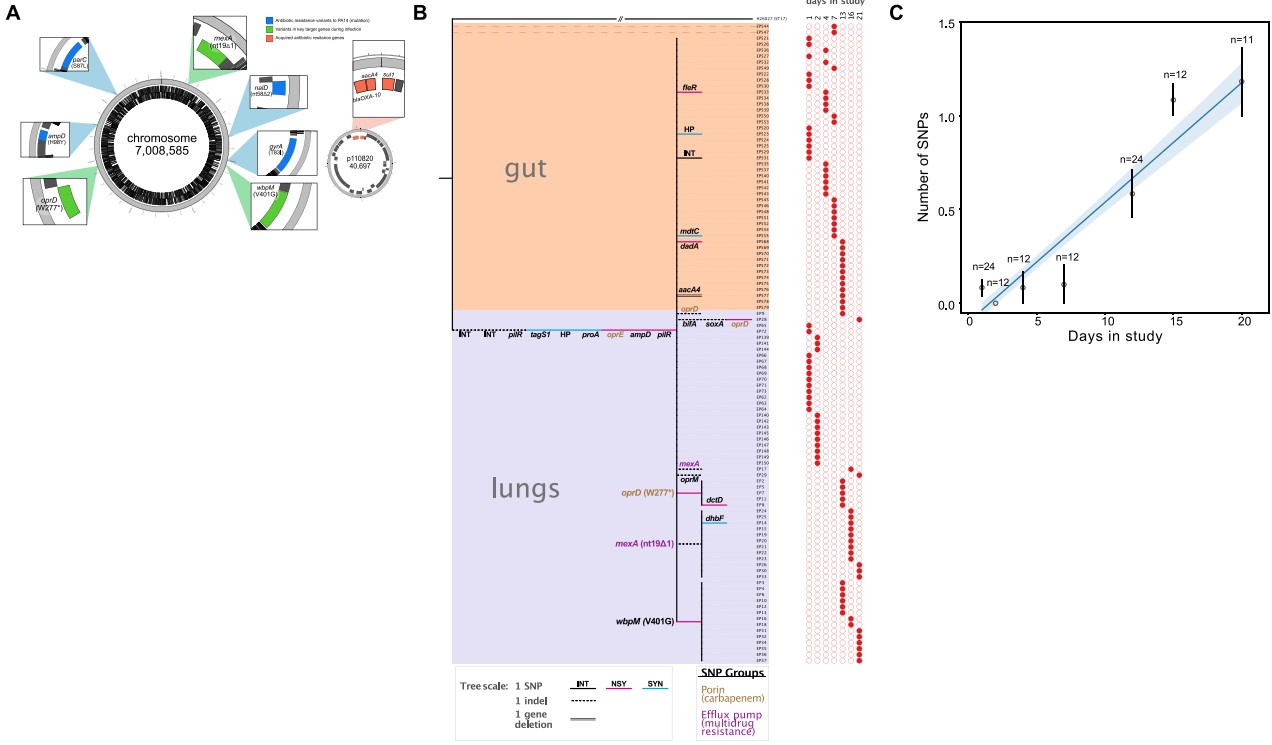

**Fig. 2 Genomic data and isolate phylogeny. A** Closed reference genome of the ST17 clone that initiated lung infection, highlighting pre-existing resistance genes and key variants acquired during infection. **B** Neighbor-joining tree showing SNPs and indels in lung and gut isolates compared to the reference genome, rooted to an ST17 outgroup genome (H26027). The tree shows intergenic (INT), synonymous (SYN) and non-synonymous mutations (NSY) SNPs and indels in coding and non-coding (INT) regions. Key mutations in *oprD*, *mexA*, and *wbpM* are highlighted. Note that 2 gut isolates lack 8 SNPs found in the reference genome and all of the other isolates. **C** Root-to-tip regression comparing genetic divergence from the reference genome (i.e., number of SNPs) with day of isolate sampling (mean +/− s.e.m; n > 10 isolates per time point). Note that this plot excludes the two outlier isolates from **B**. The solid line shows a linear regression of SNP accumulation against time (+/− 95% confidence intervals). The image shown in 2A was created by J.D-C using Circos v. 0.69 (ref. [99]) and modified using Affinity Designer [West Bridgford, UK]. Source data are provided as a Source Data file.

to the closed reference genome (Fig. 2B). Using this approach, we found a small number of chromosomal SNPs ($n = 16$) and indels ($n = 9$), most of which occurred as singletons ($n = 13$). To identify the variable genetic content among our isolates (the part of the genome found only in some isolates), we compared the genetic composition of each isolate against the gene content of all isolates and validated the potential variable genome by mapping sequencing reads to the sequences of genes in these regions. The only evidence of changes in genome composition was the deletion of a plasmid-carried *aacA4* aminoglycoside resistance cassette in a single intestinal isolate.

The genetic diversity found within this patient could reflect either (i) in situ evolutionary diversification of an ancestral bacterial clone or (ii) secondary infection by closely related strains from the ST17 outbreak in this hospital. To discriminate between these possibilities, we reconstructed the phylogeny of our isolates using a closely related ST17 genome (*P. aeruginosa* H26027) as an outgroup (Fig. 2B), and we then used root-to-tip regression to estimate the time to the most recent common ancestor (MRCA) of the isolates we sequenced (Fig. 2C). We reasoned that in situ evolutionary diversification would be associated with an MRCA within the time frame of the infection, whereas recurring infections by ST17 clones from the same outbreak would be associated with an MRCA that predated this infection. The number of SNPs per isolate was well correlated ($r^2 = 0.5$) with the day of infection (slope = 0.063 SNPs/day; s.e. = 0.0062, $t = 10.12$, $P < 0.0001$), as we would expect if all of the variants detected evolved in situ during the infection by diversification of a clonal

"ancestral strain". Strikingly, we estimated that the MRCA of the isolates occurred at approximately day 0, suggesting that the initial infection was caused by the rapid growth of a single clone after the patient was admitted to ICU and placed on mechanical ventilation. For this analysis we excluded two genetically divergent gut isolates from day 7 that lacked 8 SNPs found in the reference genome. We argue that these isolates reflect a secondary gut colonization by a distinct, closely related clone of ST17.

**Mutational adaptation in the lung.** The recovery of the pulmonary *P. aeruginosa* population following antibiotic treatment was driven by the growth of *oprD, wbpM*, and MexAB-OprM mutants descended from the ancestral strain (Fig. 3A). The small number of isolates sequenced at each time point ($n = 11$ or $12$) makes it difficult to detect subtle changes in the prevalence of different mutations over time, but two broad patterns are clear.

First, the initial recovery of the *Pseudomonas* population at day 12 was driven by the growth of *oprD* and *wbpM* mutants. Given that this diversity evolved in situ, we argue that the gap in time between the population crash of the ancestral strain and the appearance of these mutants reflects the time taken for the populations of mutants to expand from a single cell to a detectable sub-lineage of cells (minimal observed density approximately $10^2$ CFU/mL). The loss of the OprD outer membrane porin is a key mutational mechanism for meropenem resistance in *P. aeruginosa*[36], and *oprD* mutations (W277*,

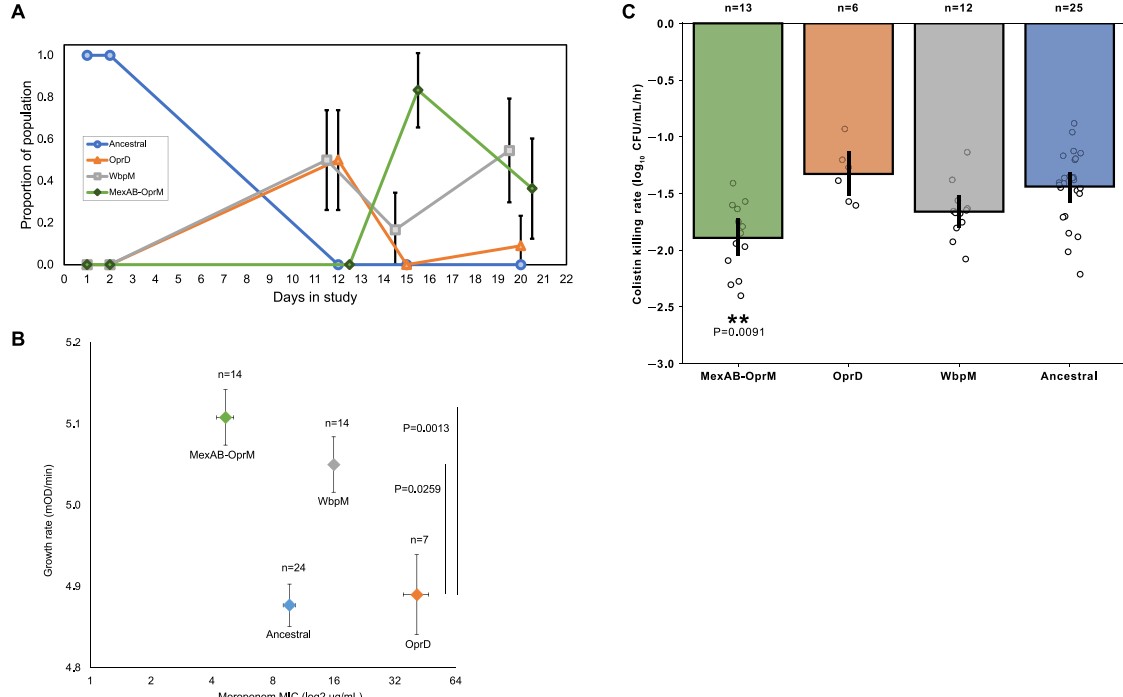

**Fig. 3 Evolutionary responses to antibiotic treatment in the lung. A** Changes in the genetic composition of the lung population over time. Plotted points show the portion of isolates of the ancestral strain and mutants that evolved in situ ($n = 11$ or 12 isolates per time point). Data points were offset for visual clarity and error bars show 90% confidence intervals in proportions calculated by the normal approximation to the binomial distribution. **B** Meropenem resistance and fitness of respiratory isolates (mean $+/-$ s.e.m; $n > 5$ isolates per group). Fitness was measured as log-phase growth rate in culture medium lacking antibiotics (10 replicates per isolate). WbpM and MexAB-OprM mutants had high fitness relative to OprD mutants, as determined by a two-tailed Dunnett's test treating *oprD* as the control group. **C** Colistin tolerance, as measured by the rate of cell killing at 2 mg/L colistin (mean $+/-$ s.e. m; $n > 5$ isolates per group). Altered colistin tolerance was only found in MexAB-OprM mutants, as determined by a two-tailed Dunnett's test treating the ancestral strain as a control group ($P = 0.0091$). Source data are provided as a Source Data file.

nt1206Δ5, W6R) were associated with large increases resistance relative to the ancestral strain (Fig. 3B; mean MIC = 41.1 mg/L; s. e.m. = 5.9; $n = 7$; Dunnett's test $P < 0.0001$). *wbpM* is part of a lipopolysaccharide biosynthesis operon and[37] mutations in this gene have previously been implicated in resistance to ß-lactam antibiotics, including meropenem[38]. Isolates with the V401G *wbpM* mutation had twofold higher levels of meropenem resistance than the ancestral strain (Fig. 3B; mean MIC = 16 mg/L; s.e.m = 0; $n = 14$; Dunnett's test, $P = 0.0045$) and we confirmed the subtle (i.e., $2x$) change in MIC associated with this gene using a PA14 *wbpM* transposon mutant.

Second, the frequency of *oprD* mutants rapidly declined, and the fall of *oprD* mutants was accompanied by the rise of mutations in MexAB-OprM, a broad-spectrum antibiotic efflux pump that was constitutively expressed in the ancestral strain[23,34]. Strikingly, we observed three independent losses of this efflux pump via frameshift mutations in either *mexA* or *oprM* (Fig. 2B), providing good evidence that the loss of this pump was adaptive. As expected, isolates with MexAB-OprM mutations had reduced resistance to meropenem relative to isolates of the ancestral strain (Fig. 3B; mean MIC = 4.85 mg/L; s.e.m = 0.45; $n = 14$; Dunnett's test; $P = 0.040$). The low meropenem resistance of MexAB-OprM mutants is intriguing, as it suggests that these mutations are likely to have arisen in sub-populations of cells of the ancestral strain that were protected by meropenem by physical barriers, such as biofilms[39], or by phenotypic resistance mechanisms, such as tolerance or persistence[40].

Antibiotic resistance mutations are usually associated with fitness costs, such as impaired growth rate and reduced competitive ability[41,42], suggesting that selection for high growth rate might have driven the demise of *oprD* mutants.

To test this hypothesis, we measured the growth rate of all the lung isolates in nutrient-rich culture medium lacking antibiotics (Fig. 3B). Although lab culture medium lacks many of the stressors encountered by pathogens during infections, this is a standardized approach for measuring the fitness of resistant mutants, and the results of these assays tend to correlate well with in vivo measures of competitive ability from animal model systems[42]. *oprD* mutants did not have reduced growth rate relative to ancestral strain, suggesting that mutations in this gene were not associated with any fitness costs per se (see also[43]). However, *oprD* mutants had low growth compared to both *wbpM* (Dunnett's test $P = 0.0259$) and MexAB-OprM (Dunnett's test $P = 0.0013$) mutants, which is consistent with the idea that the prevalence of *oprD* mutants declined due to low fitness in vivo.

The high fitness of MexAB-OprM mutants suggests that the biosynthetic burden and/or activity of this pump was costly. Bacteria often adapt to the cost of resistance through compensatory mutations that recover fitness without compromising resistance[41], and the loss of MexAB-OprM provides a clear counter-example of selection for the loss of a costly resistance determinant. Intriguingly, MexAB-OprM mutations are detected in *P. aeruginosa* from patients with cystic fibrosis[44], suggesting that selection for efflux pump inactivation is a common feature of *P. aeruginosa* infections. Given the fitness advantage enjoyed by MexAB-OprM mutants in the absence of antibiotics, it is challenging to understand why these mutants were only detected at day 16. Notably, MexAB-OprM mutants only reached detectable frequency after the end of colistin treatment, suggesting that colistin may have played a role in selection for MexAB-OprM mutations. In support of this idea, MexAB-OprM mutants

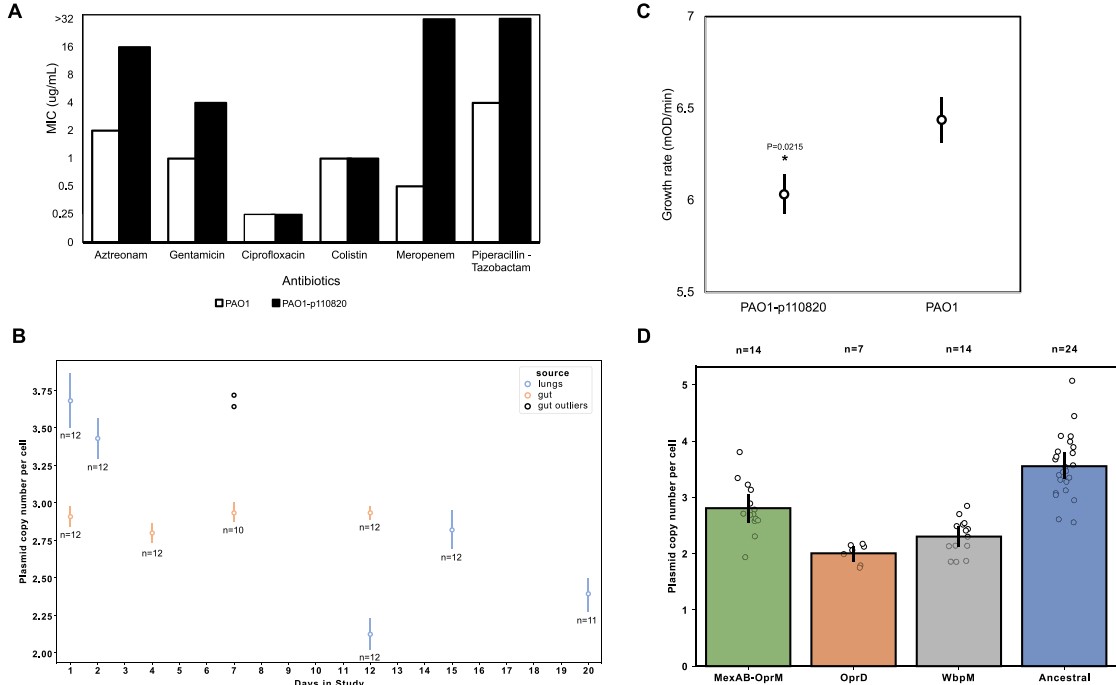

**Fig. 4 Plasmid-encoded resistance. A** Antibiotic resistance phenotypes of PA01:p110820 transformants compared to a plasmid-free PA01 control. Plasmid carriage increased resistance to meropenem and piperacillin-tazobactam to at least 32 mg/L, exceeding the EUCAST clinical breakpoints (meropenem, 8 mg/L; piperacillin-tazobactam, 16 mg/L). **B** Changes in plasmid copy number during infection. Plotted points show the mean plasmid copy number of lung (blue) and gut (beige) isolates from each time point ($+/-$ s.e.m.; $n = 10$–$12$ isolates), excluding two genetically divergent gut isolates that are plotted separately. The second wave of lung infection was associated with a reduced plasmid copy number compared to the initial infection (two-tailed $t_{57} = 8.15$, $P < 0.0001$). **C** Fitness effects of plasmid carriage were assayed by measuring the growth rate of PA01 and PA01:p110820 in antibiotic-free culture medium ($n = 11$ replicates/strain). Plasmid carriage reduced growth rate (two-tailed $t_{21} = 2.48$, $P = 0.0215$). **D** Plasmid copy number of lung isolates according to genotype (mean $+/-$ s.e.m.; $n = 7$–$24$ isolates per genotype). Source data are provided as a Source Data file.

had increased susceptibility to colistin relative to the ancestral strain (Fig. 3C; Dunnett's test $P < 0.0001$).

**Plasmid copy number evolves in response to antibiotic pressure.** All of the isolates carried a plasmid (p110820) that included an OXA-10 β-lactamase, suggesting this plasmid may have played an important role in responding to treatment with β-lactam antibiotics. To better understand the impact of this plasmid on antibiotic resistance, we transformed p110820 into the PA01 reference strain and measured antibiotic susceptibility (Fig. 4A). Although OXA-10 is generally considered to be a narrow-spectrum β-lactamase[45], plasmid carriage increased resistance to both piperacillin-tazobactam and meropenem (see also refs. [46,47]), suggesting that this plasmid played a key role in mediating the high levels of resistance to these antibiotics in the ancestral strain.

No mutations or structural variation occurred in p110820, apart from the loss of an *aacA4* aminoglycoside resistance cassette in a single intestinal isolate (Fig. 2B). However, we found subtle, but pervasive, variation in the copy number of p110820 (Fig. 4B). Copy number in intestinal isolates was ~3 per cell (mean = 2.89; s.e. = 0.054; $n = 46$), with the exception of the fact that isolates from the secondary colonization event had an elevated copy number (mean = 3.67; s.e. = 0.037; $n = 2$). The initial copy number of p110820 in pulmonary isolates was ≈20% higher than in intestinal isolates (mean = 3.55; s.e. = 0.11; $n = 24$; $t_{68} = 7.15$, $P < 0.0001$) and this was associated with a ≈20% increase in meropenem resistance in the lung isolates (mean MIC = 9.6 mg/L; s.e. = 0.67; $n = 24$) compared to the intestinal isolates (mean MIC = 8.17 mg/L; s.e. = 0.17; $n = 46$; $t_{68} = 2.75$, $P = 0.0076$),

highlighting the link between variation in plasmid copy number and antibiotic resistance.

Antibiotic resistance plasmids are often associated with fitness costs[42,48], suggesting that selection should favor reduced plasmid copy number following antibiotic treatment. Consistent with this argument, plasmid carriage reduced the growth rate of the PA01 model stain (Fig. 4C $t_{21} = 2.48$, $P = 0.0215$), and the recovery of the lung population was associated with a ≈30% reduction in plasmid copy number (Fig. 4B; mean = 2.44, s.e.n = 0.08, $n = 35$; $t_{57} = 8.15$, $P < 0.0001$). Reduced copy number was not associated with any plasmid or chromosomal mutations, suggesting that copy number declined due to selection on heterogeneity in plasmid copy generated by variation in plasmid replication and partitioning. Although low plasmid copy number is likely to have influenced the fitness and antibiotic resistance of the evolved mutants, it is difficult to quantitatively estimate this impact given that epistatic interactions between resistance plasmids and mutations are common[49]. However, it is interesting to note that plasmid copy number was high in MexAB-OprM mutants compared to *oprD* and *wbpM* mutants (Fig. 4D), suggesting that chromosomal mutations were more important determinants of fitness than plasmids copy number. In the context of antibiotic resistance, the low plasmid copy number of the evolved mutants suggests that we may have underestimated the increase in meropenem resistance provided by *wbpM* and *oprD* mutations and overestimated the loss of meropenem resistance associated with the loss of MexAB-OprM.

Although trade-offs between resistance and fitness can help to explain dynamic changes in plasmid copy number during infection, they cannot explain why plasmid copy number was

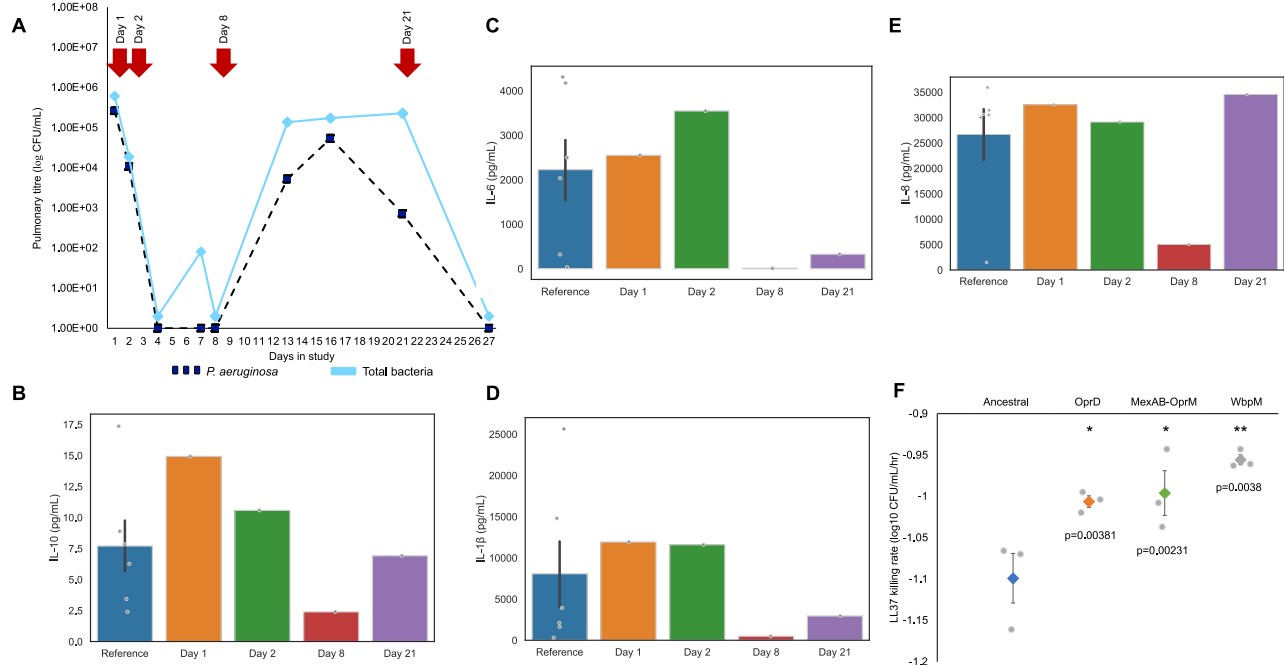

**Fig. 5 Immune responses to infection.** Cytokine concentrations were measured in samples of ETA collected over the course of infection. Panel **A** places cytokine sampling points into the context of the infection. Panels **B**–**E** show levels of cytokines that have been shown to protect against *P. aeruginosa* infection. A single measure of cytokine abundance was taken from each ETA sample due to the high reproducibility of these assays. Reference shows the abundance of cytokines in critically ill patients from the ASPIRE-ICU study who did not develop pneumonia (mean $+/-$ s.e.m; $n = 6$ patients). **F** LL-37 tolerance of lung isolates, as measured by the rate of cell death at a fixed dose of LL-37 (50 μg/mL). Plotted points show death rate of each genotype (mean $+/-$ s.e.m; $n = 3$ isolates). Increased LL-37 tolerance was only found in all mutants, as determined by a two-tailed Dunnett's test treating the ancestral strain as a control group. Source data are provided as a Source Data file.

high in the lung at the outset of the infection, prior to antibiotic treatment (i.e., on day 1). One possible explanation for this result is that a population bottleneck occurred during the initiation of the lung infection, resulting in increased copy number driven by a founder effect.

**Immunity**. Although antibiotic treatment clearly had important effects on the population dynamics and evolution of *P. aeruginosa* during lung infection, there are several features of the clinical data that antibiotic treatment alone cannot explain. First, the titer of *Pseudomonas* in the lung decreased rapidly (by >1 log) before the onset of antibiotic treatment. Second, the eventual elimination of the entire lung microbiome, including *P. aeruginosa*, was not driven by antibiotic treatment. To investigate the role of host immunity in shaping the dynamics of infection, we measured the abundance of cytokines in ETA samples taken from day 1 and 2 (initial infection), day 8 and day 23 (second wave), as shown in Fig. 5(A–E). The great advantage of this approach is that it allowed us to measure the immune response at the site of infection, instead of using a proxy measure of immunity, such as serum levels of antibodies. Importantly, the cytokines that we assayed have been shown to provide protection against *P. aeruginosa* lung infection[24,50]. Crucially, the decline of *Pseudomonas* titer during both the first wave (day 1 and 2) and second wave (day 23) coincided with high levels of expression of protective inflammatory cytokines relative to the day 8 time point, when the lung of this patient did not contain any culturable bacteria.

Neutrophils are known to play a key role in providing protection against acute *P. aeruginosa* infection in the lung[50,51]. To assay the functional consequences of elevated IL-8 (neutrophil chemoattractant) expression, we measure the resistance of bacterial isolates to LL37 and HBD-3, antimicrobial peptides

that are produced by neutrophils and lung epithelial cells[24,50,52]. All of the isolates were highly resistant to HBD-3 (350 μg/mL), but were rapidly killed by a physiologically relevant concentration of LL37 (50 μg/mL), suggesting that this host antimicrobial peptide may have played an important role in eliminating pulmonary bacteria (Fig. 5F; average LL37 killing rate $= -1.10$ $\log_{10}$ CFU/h; s.e. $= 0.018$; $n = 12$). Surprisingly, the *oprD*, *wbpM* and *mexA* mutants that appeared during the second wave of infection all showed reduced susceptibility to LL37 compared to the ancestral strain (Fig. 5F; ANOVA $F_{3,11} = 8.02$, $P = 0.0085$; all Dunnett's test $P < 0.05$). However, the magnitude of this difference was very small; for example, the time taken for LL37 to cause a 10-fold reduction in viable cell density was 54.6 min in the ancestral strain (s.e. $= 1.5$ min; $n = 3$) and 61 min in the evolved mutants (s.e. $= 0.71$ min; $n = 9$). Given the uncertainties over the effective concentrations of LL37 encountered by bacteria in the lung during this infection, it is unclear if these subtle differences in LL37 sensitivity between mutants had any biological significance during this infection.

## Discussion

Although it has long been known that antibiotic treatment can drive the rise of resistance during infections, the underlying dynamics of this process remain poorly characterized, especially during acute infections. Combining clinical data, resistance phenotyping, genomics, fitness assays and immune response profiling enabled us to produce a very high-resolution understanding of the evolutionary trajectory and drivers of antibiotic resistance during a hospital-acquired *P. aeruginosa* infection, as summarized in Fig. 6.

It is widely acknowledged that antibiotics and host immunity work in conjunction to suppress bacterial infections, but the

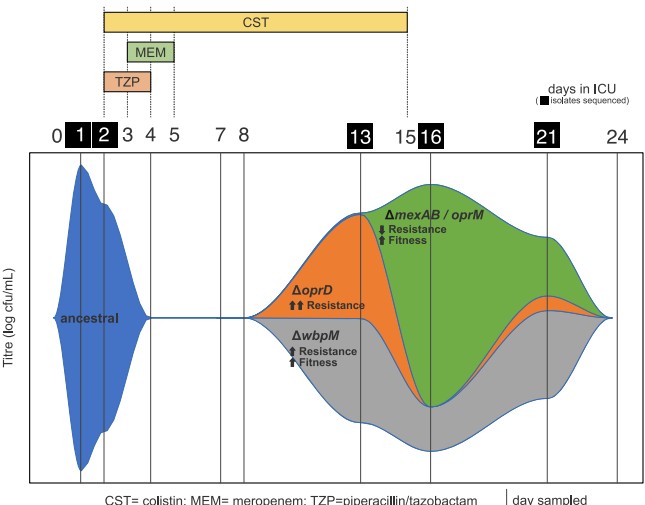

**Fig. 6 Summary of bacterial dynamics during infection.** This Muller plot summarizes changes in the density and composition of the lung population of *P. aeruginosa* during infection and it highlights the key phenotypic effects of observed mutations. This plot does not represent the reduction in the copy number of the p110820 plasmid during infection (Fig. 4C).

dynamic interplay between antibiotics, immunity and pathogens remains poorly understood[7–9]. In this case, host immunity was able to reduce pathogen density by at least 1 log prior to the onset of antibiotic treatment. The toxicity of antibiotics towards bacteria is greatest at low bacterial cell density[53], suggesting that early (i.e., day 1–2) immunity-mediated suppression of bacterial population density may have increased the efficacy of meropenem and colistin treatment (day 3–4). Combination therapy with meropenem and colistin contributed to the successful suppression of initial infection, but the *Pseudomonas* population recovered due to the successful outgrowth of meropenem resistant mutants, highlighting the incredible ability of *P. aeruginosa* to evolve mutational resistance to clinically important carbapenem antibiotics[54–56].

Theoretical considerations suggest that the immune-mediated suppression of bacterial population density is likely to have constrained the evolutionary response to antibiotic treatment[57]. First, reducing population density must have decreased the absolute number of antibiotic resistant mutants that were present at the time of meropenem-colistin treatment. Reducing pathogen density prior to antibiotic treatment may have also decreased the likelihood of successful outgrowth of resistant mutants by increasing the effective exposure of resistant cells to antibiotics[58], an effect that is likely to be particularly important for mutations such as *wbpM* that lead to small increases in resistance. While these constraints were not able to prevent the evolution of antibiotic resistance, there are good reasons for thinking that host immunity reduced the number of resistant mutants that were able to successfully grow following antibiotic treatment. This, in turn, is likely to have (i) increased the lag time between antibiotic treatment and the detectable recovery of the *Pseudomonas* population and (ii) decreased the diversity resistant mutants in the population following recovery.

One of the key principles of evolutionary models of resistance is that fitness costs generate selection against resistance following antibiotic treatment, leading to the loss of resistance[41,59]. In this case, relaxed antibiotic pressure drove the decline of high resistance/low fitness *oprD* mutants and the spread of mutations that inactivated a costly efflux pump. The copy number of the costly p110820 plasmid also declined following treatment, providing good evidence of selection on non-mutational variation in

plasmid copy number. Variation in plasmid copy number arises due to inherent variability in plasmid replication and partitioning, suggesting that altered plasmid copy number may be a very general, and underappreciated, evolutionary response to antibiotic treatment during infections (see also refs. [60–62]). While these examples highlight the ability of selection to drive the loss of resistance, it is important to emphasize that meropenem resistance was maintained during the second wave of infection due to the stability of the high fitness *wbpM* mutant. Our results suggest that the host immune response ultimately suppressed the second wave of infection, thereby limiting the potential for onwards transmission of *wbpM*. Crucially, these results show that selection and host immunity interact to drive the loss of resistance following treatment.

One important challenge for future work will be to investigate interactions between immunity and resistance in greater depth. In the first place, chemical interactions between antibiotics and host immunity effectors (i.e., synergy or antagonism) may modulate the efficacy of antibiotic treatment and its associated selective pressures. In this case, colistin has been shown to suppress the inflammatory response[63], suggesting that continued colistin treatment may have delayed the suppression of the second wave of infection. Antibiotic resistance mutations also alter resistance to host antimicrobial peptides (AMPs)[64], suggesting immunity may play an important role in fitness of resistant mutants. For instance, increased resistance to host AMPs may have contributed to the success of the mutants that we observed. An associated challenge will be to understand the role of the microbial community in the dynamics of resistance. Competition between bacterial species is common[65], suggesting that lung microbiome may have limited the success of meropenem resistant *P. aeruginosa* during the second wave of infection. At the same time, it is conceivable that the microbiome has indirect effects on resistance, such as host immunomodulation[66], that could either promote or prevent the growth of resistant mutants.

It is common for pathogenic bacteria to inhabit multiple anatomical sites in the body, and this has the potential to generate within-host variation in antibiotic exposure and the selective pressures that this generates[67]. Although *P. aeruginosa* is primarily considered to be an opportunistic respiratory pathogen, this bacterium is also capable of colonizing the gut. Antibiotic treatment has profound impacts on the gut microbiome, suggesting that the gut is likely to be a "hot-spot" for the evolution of resistance[68,69]. However, we found no evidence of clinical or evolutionary responses to meropenem treatment in gut, and this can be explained by the poor penetrance of meropenem into the gut lumen[70] and the low toxicity of meropenem under anaerobic conditions. Given this, our data suggests that the gut provided *P. aeruginosa* with an effective refuge against antibiotic treatment. However, the importance of gut colonization in the infection biology of *P. aeruginosa* remains unclear. On the one hand, gut colonization may simply be a dead-end, as appears to have been the case in this patient. Alternatively, gut infections that are protected from antibiotic treatment may act as a reservoir that can establish infections in new anatomical locations (i.e., lung, blood stream) or in new hosts. Hopefully, future studies will resolve this issue by estimating the importance of gut populations to the transmission of *P. aeruginosa*.

Our study was able to capture the evolutionary responses of a pathogen population to antibiotic treatment by characterizing the genetic and phenotypic diversity present in longitudinal samples taken from a single patient. The key insight is that natural selection and host immunity interact to drive the incredibly rapid rise, and fall, of resistance during short-term infections. Previous work that has characterized the evolutionary dynamics in patients has relied largely on long-term sampling of chronic

infections[10–16] or taking multiple samples from a single time point[71,72]. Although our study focused on a single patient, our findings highlight that infrequent sampling of pathogen populations may underestimate the rate of evolution of resistance because of the fast turnover of resistant lineages following treatment. Furthermore, capturing host immune responses allowed us to better understand the drivers of resistance, and there is a clear need to better understand both direct and indirect interactions between immunity and resistance. Hopefully, future studies using high-resolution sampling across multiple patients will help to resolve this.

## Methods

**Clinical data.** The patient was recruited as part of an observational, prospective, multicentre European epidemiological cohort study, ASPIRE-ICU (The Advanced understanding of *Staphylococcus aureus* and *Pseudomonas aeruginosa* Infections in Europe–Intensive Care Units, NCT02413242 ClinicalTrials.gov)[27]. The intervention was standard of care, and the research protocol was approved by the Andalusian Biomedical Research Ethincs Coordinating Committee (CCEIBA). An agreed legal representative of the participant gave written informed consent, according to CARE guidelines and in compliance with the Declaration of Helsinki principles. ASPIRE-ICU enrolled subjects who were mechanically ventilated at ICU admission and with an expected length of hospital stay ≥48 h. An assessment of four clinical criteria to establish a clinical diagnosis of ICU pneumonia (e.g., new blood culture drawn, new antibiotic use, new radiologic evidence, reason to suspect pneumonia) was performed daily; in case of at least one positive parameter, a combination of objective major and minor criteria was assessed to categorize subjects as having protocol pneumonia or not[27]. Data on antibiotic use in the two weeks preceding ICU admission and during the ICU stay were reported. During ICU stay, study samples (e.g., lower respiratory tract samples and peri-anal swabs) were obtained three times weekly in the first week, two times weekly in the three following weeks and on the day of diagnosis of protocol pneumonia and seven days after it.

**Sample collection and isolation.** The respiratory samples and peri-anal swabs used in this study were collected within the ASPIRE-ICU study and are from a single patient at a Spanish hospital[27]. Respiratory samples were collected by endotracheal aspiration on the following visit days: 1 (the day of informed consent, 72 h after ICU admission), 4, 7, and twice weekly for 30 days or until ICU discharge. In this case: day 10, 13, 16, 21, 23, 27. From patients who were diagnosed with pneumonia, additional respiratory samples were collected at the day of diagnosis and 7 days post-infection: day 2 and 8. Peri-anal swabs in skimmed milk medium and untreated respiratory samples were stored at −80 °C until shipment to the Central lab at the University of Antwerp and until further analysis. Semi-quantitative culture of peri-anal swabs was performed by inoculating the swabs directly on CHROMID *P. aeruginosa* Agar (BioMérieux, France) and blood agar (BBL®Columbia II Agar Base (BD Diagnostics, USA) supplemented with 5% defibrinated horse blood (TCS Bioscience, UK)). After incubation of 24 h at 37 °C, the growth of *P. aeruginosa* was evaluated in four quadrants.

Patient endotracheal aspirate (ETA) samples were blended (30,000 rpm, probe size 8 mm, steps of 10 s, max 60 s in total), diluted 1:1 v/v with Lysomucil (10% Acetylcysteine solution) (Zambon S.A, Belgium) and incubated for 30 min at 37 °C with 10 s vortexing every 15 min. Thereafter, quantitative culture was performed by inoculating 10-fold dilutions on CHROMID *P. aeruginosa* Agar and blood agar using spiral plater EddyJet (IUL, Spain). Plates were incubated at 37 °C for 24 h and CFU/mL was calculated. Plates without growth were further incubated for 48 h and 72 h. Matrix-Assisted Laser Desorption Ionization-Time of Flight Mass Spectrometry (MALDI-TOF MS) was used to identify 12 *P. aeruginosa* colonies per sample, which were stored at −80 °C until further use. One respiratory isolate was subsequently identified as *S. epidermidis* from whole-genome sequencing and this isolate was excluded from all analysis. MALDI-TOF was also used to identify bacterial colonies to species level on blood agar plate and the rank-order abundance of species on these plates was recorded.

**Resistance phenotyping.** All isolates were grown from glycerol stocks on Luria-Bertani (LB) Miller Agar plates overnight at 37 °C. Single colonies were then inoculated into LB Miller broth for 18–20 h overnight growth at 37 °C with shaking at 225 rpm. Overnight suspensions were serial diluted to ~5 × 10^5 CFU/mL. Resistance phenotyping was carried out as minimum inhibitory concentration (MIC) testing via broth microdilution as defined by EUCAST recommendations[73,74], with the alteration of LB Miller broth for growth media and the use of *P. aeruginosa* PAO1 as a reference strain[75]. We defined growth inhibition as OD$_{595}$ < 0.200 and we calculated the MIC of each isolate as the median MIC score from three biologically independent assays of each isolate. We used a one-way ANOVA that included a main effect of genotype (ancestral, oprD, wbpM or MexAB-OprM) to test for variation in meropenem resistance. We then used a Dunnett's test to compare the evolved mutants against the ancestral strain.

Statistical analysis was performed using JMP v.12. A checkerboard assay was used to test the combined effect of meropenem and colistin. We used the same methods as above, using all possible combinations of a log$_2$ dilution series of meropenem (0–64 mg/L) and colistin (0–2 mg/L). This assay was performed using four independent isolates of the ancestral strain.

**Colistin tolerance assay.** All isolates were grown from glycerol stocks on LB Miller Agar plates overnight at 37 °C. Each culture was grown from a single, randomly selected colony, inoculated into 200ul of LB Miller and grown over 18–20 h at 37 °C with shaking at 225RMP. Overnight cultures were diluted in phosphate saline buffer to a final concentration of ~1 × 10^6 CFU/mL, further verified by total viable count, and grown in 200 μl LB Miller with or without the addition of 2 mg/L colistin. To avoid colistin carry-over, cultures were diluted at least 10-fold and plated on LB Miller agar after 0, 1, 2, 4, or 8 h. This assay was carried out using a randomized block experimental design, and we analyzed five replicates of 11 randomly selected isolates and five replicates of a PA01 control in each block. The order and position of each isolate on the experimental plates was selected through randomization, using "sample" command without replacement in R[76]. We used linear regression of mean log viable cell titer against time to calculate a death rate for each isolate (typically this involved data from 0 to 2 h of incubation). In no case did we observe bi-phasic killing kinetics. We used a one-way ANOVA that included main effects of experimental block and genotype (ancestral, oprD, wbpM or MexAB-OprM) to test for variation in colistin tolerance scores between mutants. We then used a Dunnett's test to compare the evolved mutants against the ancestral strain. Statistical analysis was performed using JMP v.12.

**Anaerobic meropenem resistance assay.** Six isolates of the ancestral strain were grown from glycerol stocks on Luria-Bertani (LB) Miller Agar plates overnight at 37 °C. Single colonies were then inoculated into LB Miller broth supplemented with meropenem at increasing twofold concentrations (1–64 μg/mL) and grown in an anaerobic jar (Thermo Scientific Oxoid AnaeroJar™ with anaerobic gas generating sachet) for 72 h at 37 °C. MIC was calculated as MIC$_{50}$ (a 50% reduction) in OD$_{595}$ and compared to the same calculation under standard aerobic conditions. We calculated an MIC$_{50}$, rather than a conventional MIC, due to the low growth of the ancestral strain under anaerobic conditions.

**Characterization of the *wbpM* mutant.** In order to determine the effect of the *wbpM* in resistance, meropenem MICs were determined by EUCAST broth microdilution in triplicate experiments for wild-type reference strain PA14 and its *wbpM* isogenic knock out derivative obtained from an available transposon mutant library[77].

**Gene expression.** The levels of expression of *ampC* and *mexB* were determined by real-time reverse transcription (RT)-PCR[78,79]. Briefly, isolates were grown in 10 mL of LB broth at 37 °C and 180 rpm to the late log-phase (optical density at 600 nm [OD600] of 1) and collected by centrifugation. Total RNA was isolated by using the RNeasy minikit (Qiagen), dissolved in water, and treated with 2 U of Turbo DNase (Ambion) for 30 min at 37 °C to remove residual contaminating DNA. A 50 ng sample of purified RNA was then used for one-step reverse transcription and real-time PCR amplification using the QuantiTect SYBR green RT-PCR kit (Qiagen) with a Bio-Rad instrument (CFX Connect Real-Time System). Primers (Supplementary Table 1) were used for the amplification of *ampC*, *mexB*, and *rpsL* (used as a reference to normalize the relative amount of messenger RNA (mRNA)). To calculate *ampC* and *mexB* mRNA expression levels compared to PAO1 the following formula was applied:

$$2^{\Delta Ct}, \text{ being } \Delta Ct = Ct_{PAO1,target} - [Ct_{Isolate,target} + (Ct_{PAO1,rpsL} - Ct_{Isolate,rpsL})]$$

Isolates were considered positive for *ampC* overexpression when the corresponding mRNA level was at least 10-fold higher than that of PAO1[80]. Likewise, isolates were considered positive for *mexB* overexpression when the corresponding mRNA level was at least threefold higher than that of PAO1[80]. Mean values (±standard deviations) of mRNA levels obtained in at least two independent duplicate experiments were considered for each isolate (Source Data file).

**Long-read sequence analysis.** Four isolates were sequenced with the Pacific Biosciences platform using single molecule chemistry on a SMRT DNA sequencing system. Coverage ranged from 122X to 171X. Resulting sequencing reads were assembled using canu v. 1,8 indicating a genome size of 7 Mb and using raw error rate of 0.300, corrected error rate of 0.045, minimum read length of 1000 bases, and minimum overlap length of 500[81]. Canu assemblies were circularized using circlator v.1.5.5 testing kmer sizes 77, 87, 97, 107,117, and 127, minimum merge length of 4000, minimum merge identity of 0.95, and minimum contig length of 2000[82].

**Illumina sequence analysis.** All isolates were sequenced in the MiSeq or NextSeq illumina platforms yielding a sequencing coverage of 69X–134X. Raw reads were quality controlled with the ILLUMINACLIP (2:30:10) and SLIDINGWINDOW

(4:15) in trimmomatic v. 0.39[83]. Quality controlled reads were assembled for each isolate with SPAdes v. 3.13.1 with default parameters[84]. These assemblies were further polished using pilon v. 1.23 with minimum number of flank bases of 10, gap margin of 100,000, and kmer size of 47[85]. Resulting contigs were annotated based on the *P. aeruginosa* strain UCBPP-PA14[86] in prokka v. 1.14.0[87].

**Variant calling**. To identify pre-existing resistance mutations that were present at the start of the infection reads for each of the isolates were mapped to the *P. aeruginosa* PAO1 reference genome (GenBank accession: NC_002516.2) with Bowtie 2 v2.2.4[88] and pileup and raw files were obtained by using SAMtools v0.1.16[89] and PicardTools v1.140[90], using the Genome Analysis Toolkit (GATK) v3.4.46 for realignment around InDels[91]. From the obtained raw files, SNPs were extracted if they met the following criteria: a quality score (Phred-scaled probability of the samples reads being a homozygous reference) of at least 50, a root-mean-square (RMS) mapping quality of at least 25 and a coverage depth of at least three reads; excluding all ambiguous variants. As well, MicroInDels were extracted from the total pileup files when meeting the following: a quality score of at least 500, a RMS mapping quality of at least 25 and support from at least one-fifth of the covering reads. Filtered files were eventually annotated with SnpEff v4.2 and SNPs and InDels located in a set of genes known to be involved in *P. aeruginosa* chromosomal antibiotic resistance were extracted[92–94].

To identify mutations and gene gain/loss during the infection, short-length sequencing reads from each isolate were mapped to each of the four long-read de novo assemblies with bwa v. 0.7.17 using the BWA-MEM algorithm[95]. Preliminary SNPs were identified with SAMtools and BCFtools v. 1.9[89]. Low-quality SNPs were filtered out using a two-step SNP calling pipeline, which first identified potential SNPs using the following criteria: 1. Variant Phred quality score of 30 or higher, 2. At least 150 bases away from contig edge or indel, and 3. 20 or more sequencing reads covering the potential SNP position[10]. In the second step, each preliminary SNP was reviewed for evidence of support for the reference or the variant base; at least 80% of reads of Phred quality score of 25 or higher were required to support the final call. An ambiguous call was defined as one with not enough support for the reference or the variant, and, in total, only one non-phylogenetically informative SNP position had ambiguous calls. Indels were identified by the overlap between the HaplotypeCaller of GATK v. 4.1.3.0[96] and breseq v. 0.34.0[97]. The variable genome was surveyed using GenAPI v. 1.0[98] based on the prokka annotation of the short-read de novo assemblies. The presence or absence of genes in the potential variable genome was reviewed by mapping the sequencing reads to the respective genes with BWA v.0.7.17[92–94].

**Growth rate assays**. All isolates were grown from glycerol stocks on LB Miller Agar plates overnight at 37 °C. Single colonies were then inoculated into LB Miller broth for 18–20 h overnight growth at 37 °C with shaking at 225 rpm. Overnight suspensions were diluted to an $OD_{595}$ of ~0.05 and placed within the inner 60 wells of a 96-well plate equipped with a lid. To assess growth rate, isolates were then grown in LB Miller broth at 37 °C and optical density (OD595nm) measurements were taken at 10-min intervals in a BioTek Synergy 2 microplate reader set to moderate continuous shaking. Growth rate was calculated as the maximum slope of OD versus time over an interval of ten consecutive readings, and we visually inspected plots to confirm that this captured log-phase growth rate. We measured the growth rate of ten replicate cultures of all 60 pulmonary isolates. We used an ANOVA to test for variation in growth rate using a model that included a main term of genotype (ancestral, oprD, wbpM or MexAB-OprM) and a nested effect of isolate (i.e., isolates nested within genotypes). Dunnett's test was then used to test for differences between genotypes compared to *oprD* mutants. Statistical analysis was performed in JMPv12.

**Plasmid transformation and characterization**. The p110820 plasmid was extracted from an ancestral lung isolate (Thermo Scientific Plasmid Miniprep Kit), transformed into a wild-type *P. aeruginosa* PAO1 background[75] via electroporation using a MicroPulsar Electroporator (Bio-Rad), and successful transformants (PAO1-p110820) were confirmed via sanger sequencing. Growth rate assays and resistance phenotyping of PAO1-p110820 and a PAO1 control were performed as described above, with three replicates per strain for MIC assays and 11 replicates per strain for growth rate assays. We tested for a difference in growth rate between PA01 and PA01:p110820 by *t*-test using JMPv.12.

**Cytokine profiling**. After ETA was blended, 0.5 g of the sample was diluted 1:1 with Sputolysin (Merck, Overijse, Belgium), vortexed and incubated at room temperature for 15 min. Samples were then centrifuged for 5 min at $2000 \times g$ at room temperature. Supernatant was stored at −80 °C until further processing. Cytokine levels were measured with the Mesoscale Discovery platform (Rockville, MD, USA) following the manufacturer's instructions. In brief, the plate was coated with capturing antibodies for 1 h with shaking incubation at room temperature followed by washing off the plate. Samples were loaded and incubated for 1 h, after which the plate was washed and incubated with detection antibodies. A final wash was performed and MSD reading buffer 2x was applied before reading the plate in the QuickPlex SQ 120 (Rockville, MD, USA).

**LL-37 tolerance assay**. A time-kill curve study was carried out to measure the tolerance of ancestral and mutant strains (*oprD*, *wbpM* and *mexA*) to human cathelicidin peptide LL-37. Tolerance was measured by determining the change in bacterial population size upon exposure to a lethal concentration of LL-37. Bacterial strains (three biological replicates/strain) were grown in Mueller-Hinton broth (MHB) medium for overnight at 30 °C with shaking at 250 rpm. Overnight cultures were inoculated into 5X-diluted MHB medium containing LL-37 (50 μg/mL) at an initial density of ~$3 \times 10^5$ CFU/mL. A diluted growth medium was used because a high concentration of salts interferes with the activity of antimicrobial peptides. Cultures containing LL-37 were incubated at 37 °C and samples were taken at multiple time points (0, 30, 60, 120, and 180 min postexposure) from each culture (three replicate cultures per strain). Samples were diluted in PBS and spread on LB agar plates and colonies were counted after overnight incubation at 37 °C. The LL-37 killing rate for each culture was calculated from a linear regression of $\log_{10}$ viable cell titer against time. We did not observe any bi-phasic killing curves. We used a one-way ANOVA that included a main effect of genotype (ancestral, OprD, WbpM and MexAB-OprM) to test for variation in LL37 resistance. We then used a Dunnett's test to compare the evolved mutants against the ancestral strain. Statistical analysis was carried out in JMP v.12.

**Reporting summary**. Further information on research design is available in the Nature Research Reporting Summary linked to this article.

## Data availability
Data is available from figshare [https://doi.org/10.6084/m9.figshare.14219129.v1]. All clinical data analyzed for this patient as part of the study are included in this article. Isolates can be obtained from the corresponding author for research use via an MTA subject to permission from the ASPIRE research committee. All sequencing data has been deposited on the NCBI short-read archive ("PRJNA667268") and all data on isolates can be found at "SRR12772624", "SRR12772625", "SRR12772626", "SRR12772627", "SRR12772628", "SRR12772629", "SRR12772630", "SRR12772631", "SRR12772632", "SRR12772633", "SRR12772634", "SRR12772635", "SRR12772636", "SRR12772637", "SRR12772638", "SRR12772639", "SRR12772640", "SRR12772641", "SRR12772642", "SRR12772643", "SRR12772644", "SRR12772645", "SRR12772646", "SRR12772647", "SRR12772648", "SRR12772649", "SRR12772650", "SRR12772651", "SRR12772652", "SRR12772653", "SRR12772654", "SRR12772655", "SRR12772656", "SRR12772657", "SRR12772658", "SRR12772659", "SRR12772660", "SRR12772661", "SRR12772662", "SRR12772663", "SRR12772664", "SRR12772665", "SRR12772666", "SRR12772667", "SRR12772668", "SRR12772669", "SRR12772670", "SRR12772671", "SRR12772672", "SRR12772673", "SRR12772674", "SRR12772675", "SRR12772676", "SRR12772677", "SRR12772678", "SRR12772679", "SRR12772680", "SRR12772681", "SRR12772682", "SRR12772683", "SRR12772684", "SRR12772685", "SRR12772686", "SRR12772687", "SRR12772688", "SRR12772689", "SRR12772690", "SRR12772691", "SRR12772692", "SRR12772693", "SRR12772694", "SRR12772695", "SRR12772696", "SRR12772697", "SRR12772698", "SRR12772699", "SRR12772700", "SRR12772701", "SRR12772702", "SRR12772703", "SRR12772704", "SRR12772705", "SRR12772706", "SRR12772707", "SRR12772708", "SRR12772709", "SRR12772710", "SRR12772711", "SRR12772712", "SRR12772713", "SRR12772714", "SRR12772715", "SRR12772716", "SRR12772717", "SRR12772718", "SRR12772719", "SRR12772720", "SRR12772721", "SRR12772722", "SRR12772723", "SRR12772724", "SRR12772725", "SRR12772726", "SRR12772727", "SRR12772728", "SRR12772729", "SRR12772730". Source data are provided with this paper.

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

## Acknowledgements

This research was supported by Wellcome Trust Grant (106918/Z/15/Z) and the Innovative Medicines Initiative Joint Undertaking under COMBACTE-MAGNET (Combatting Bacterial Resistance in Europe-Molecules against Gram-negative Infections, grant agreement no. 115737) and COMBACTE-NET (Combatting Bacterial Resistance in Europe-Networks, grant agreement no. 115523), resources of which are composed of financial contribution from the European Union's Seventh Framework Program (FP7/ 2007-2013) and EFPIA companies' in kind contribution. We thank the Oxford Genomics Center (funded by Wellcome Trust Grant 203141/Z/16/Z) for the generation and initial processing of Illumina sequence data.

## Author contributions

R.W., J.D.C., N.K., F.H.R.d.W., P.J., A.Q., G.T., T.V.d.S., F.F.-.C., A.A., J.H., E.d.B.-T., C.L.-C., B.B.X., C.R., L.T., C.L., contributed to data acquisition and analysis. F.S., O.A., A.R., H.G., J.K., S.K.-S., A.O., S.M.-K., and C.M. contributed to project conception and study design. R.W., J.D.C., A.O., S.M.-.K., and C.M. wrote and revised the manuscript.

## Competing interests

The authors declare no competing interests.
