## [Peer Review File · Nature Communications]

REVIEWER COMMENTS

Reviewer #1 (Remarks to the Author):

To understand population responses to antibiotic therapy, the authors sampled *P. aeruginosa* isolates (>100 from the infected respiratory site and from the gut) from one patient undergoing antibiotic therapy over 27 days and used WGS to characterize genetic changes that were associated with a raise of the bacterial population in the respiratory tract after discontinuation of antibiotic therapy.

The authors convincingly demonstrate that the raise of the population following meropenem therapy was associated with the development of resistance (via the acquisition of mutations in *oprD* and *wbpM*). However the *oprD* mutants, which exhibited elevated resistance levels, were later replaced by mutants that showed inactivated multi-drug efflux pumps (anti-resistance phenotype). Their raise was probably due to their increased fitness as compared to the ancestral strain, which was constitutively overexpressing the efflux pump. The other (weaker) resistance determinant (mutation in *wbpM*), was not replaced. However, this mutant exhibited a fitness advantage over the ancestral strain.

The study is well-designed and provides a high-resolution characterization of bacterial responses to antibiotic treatment. It shows that antibiotic therapy selects for resistance conferring mutations and once discontinued, mutations that increase fitness predominate. Interestingly, these changes in the population was exclusively seen at the infected site (and not in the colonized gut).

I am not absolutely convinced that the results of this study “suggest that selection for anti-resistance mutations may be a common feature of *P. aeruginosa* infections” (line 275). There seems to be only a selection for anti-resistance, if the resistance conferring mutation was associated with a fitness burden – as in the case of *mexAB* pump overexpression. No selection for anti-resistance was observed for the *wbpM* mutation.

Specific comments:

Can the finding of a lack of an evolutionary response in the gut be correlated to the different population size in the gut? And can this be experimentally shown by e.g. treating large and small bottleneck populations of the ancestral isolate with meropenem in vitro?

Did the authors find other gain-of fitness mutations (in addition to the efflux pump mutations) that became apparent in the late isolates?

The greatest fitness increase should be observed in a *mexAB-OprM* / *wbpM* double mutant (pro and anti-resistant). Did the authors observe co-occurrence of the two mutations in single clinical isolates and what is the effect on fitness?

Reviewer #2 (Remarks to the Author):

Thank you for asking me to review this interesting study by Wheatley et al, in which the authors characterise the population and genetic changes in cultured *Pseudomonas aeruginosa* isolates with a focus on mechanisms of antimicrobial resistance within a single patient admitted to critical care with a ventilator-associated pneumonia (VAP). In total they analysed n=107 *P. aeruginosa* isolates (n=59 from endotracheal aspirate samples; n=48 from peri-anal swabs) at 7 timepoints over a 21-day timeframe, using a combination of whole genome sequencing (WGS), growth rate assays to assess fitness, and antimicrobial susceptibility testing to assess resistance phenotypes.

They conclude: (i) bacterial populations in acute *P. aeruginosa* infections in the lung fluctuate rapidly in response to antimicrobial selection pressures; (ii) a short-course (2 days) of meropenem treatment drove the selection of deltaoprD (W829*) and deltawbpM (V1201G) mutants captured 8 days later (at D13); and (iii) subsequent replacement of oprD mutants with MexAB-OprM efflux pump mutants with relatively higher growth rates and decreased colistin resistance then occurred, once a 13-day treatment course of colistin had been completed at D14.

Overall, developing a better understanding of the real-life selection and persistence of resistance in bacterial populations is of great scientific and clinical interest, the design of the study is elegant, and the manuscript is well-written. However, whilst I appreciate that such detailed analyses of genotypic and phenotypic diversity are difficult and resource-intensive, caution is required in drawing generalizable conclusions from a study of these changes in a single individual (i.e. this is effectively a study of n=1), particularly when ancestral populations of the same clone in the GI tract of the same patient remained largely unaffected.

A consideration of other potentially relevant factors should perhaps be mentioned - for example:

- Antibiotic distributions and contributions in the site of infection being cultured (e.g. thought to be low for lung/epithelial lining fluid when IV colistin is administered <https://www.ncbi.nlm.nih.gov/pmc/articles/PMC3224467/>), which mean that it might be hard to interpret the likely in vivo impact of antimicrobials on these strains in the context of only in vitro MICs?
- How sample/sampling variability might impact on the accuracy of bacterial titre measurements?
- How characterising the diversity amongst small numbers of isolates when compared with the vast number of isolates present (~9-12 isolates at each sampling event) can then reliably be used to

extrapolate to determining the distributions of these mutants at the bacterial population-level. My understanding is that we cannot be sure how many colonies you need to sequence to accurately reflect the distribution of various strains in a sample, but it would probably be more than 12, especially in the GI tract?

- What role the human immune response might have in the selection/adaptation of various sub-strains?
- Whether differential gene expression might have a role to play?
- What interaction might be occurring between the *Pseudomonas* sub-strains and other bacteria (both cultured and uncultured) that might affect selection? The culture results in Fig.1A, suggest that beyond D13 other non-*Pseudomonas* culturable bacteria are clearly present?

Metagenomics might be a complementary, additional way to look more broadly for strain-level diversity that is not identified by picking a small subset of colonies for genomic evaluation (e.g. by using strain-aware metagenomic profilers, such as <https://pubmed.ncbi.nlm.nih.gov/28167665/>). This could be done either on sweeps of cultured selected growth, and/or on unselected growth to evaluate wider bacterial species diversity in samples. Transcriptomics might be a useful way to understand genome-wide differences in gene expression that may be impacting on selection/adaptation that are not reflected in DNA sequencing analyses.

In addition to the above, please consider the following comments:

1. I think the plan was to analyse 12 isolates per sample, and nine samples were evaluated (n=5 ETA, 4 peri-rectal) - why then are there only 107/108 isolates sequenced (seems to affect lung sample at D21)? Was there a sequencing failure? Please explain and include a short statement as to why this isolate was excluded.

2. Ancestral *P. aeruginosa* populations in the lung appear to be declining before piperacillin-tazobactam and colistin treatment starts (decrease in counts from 1×10^6 to 1×10^4 CFU/mL - Fig. 1A from days 1>2) - what do the authors propose is the cause of this?

3. Why is there a delay in the emergence of meropenem-R mutants if meropenem is thought to be the main selection pressure driving this emergence (meropenem administration stops on D4, mutants appear to emerge after D8), and why should meropenem represent a significant selection pressure for these mutants in the first place given that the ancestral isolates already have resistant MICs at baseline (>8 mg/L, Fig 1C)? I can't find any evidence of a post-antibiotic effect for meropenem that lasts beyond several hours.

4. Given wbpM's proposed role in LPS synthesis (as in ref 26 in the manuscript), and colistin's mode of action (interaction with LPS and then destabilisation of the cell membrane), isn't sub-therapeutic colistin as a selection pressure driving the emergence of the wbpM mutants a possible alternative explanation? Colistin susceptibility testing remains a slightly unreliable business, we are not clear on the in vivo drug concentrations achieved in the lung (e.g. <https://academic.oup.com/cid/advance-article/doi/10.1093/cid/ciaa121/5735218>), and these mutant populations emerged during colistin (not meropenem) monotherapy. Notably, the mexAB-oprM mutants (significantly less colistin-tolerant) only emerge after colistin therapy has stopped, and at this point briefly out-compete the wbpM mutant, probably because of faster growth rates. Despite the in vitro colistin testing findings, is this really population adaptation in response to meropenem and colistin therapy, and not just meropenem administration?

5. The authors mention this briefly but don't hypothesise why/answer the question in the text - if antibiotic selection pressures are the critical driving force, why was the same clonal ancestral gastrointestinal population of *P. aeruginosa* in this patient unaffected by the same antibiotic exposures?

6. Lines 171-176 - With respect to the accessory genome analysis (based here on using Prokka annotations and GenAPI to identify presence/absence of annotated genes) - would it be worthwhile undertaking a k-mer based evaluation on non-core reads to determine if there is any variability that is not captured by looking only at annotated regions (e.g. in promoters etc.)? Previous analyses e.g. <https://academic.oup.com/gbe/article/11/5/1385/5454722> suggest that mutations in intergenic regions (i.e. potentially more likely to be un-annotated) might be very important in adaptation and "are directly responsible for the evolution of important pathogenic phenotypes including antibiotic sensitivity".

7. Lines 189-192 - the authors describe 12 isolates that were cultured on D13 from the ETA samples. Please describe/mention the other genotype (indel in oprD from what I can see in Fig.2B)?

8. Was there any evidence that gene duplication might be playing a role in adaptation (could perhaps be evaluated from looking at read coverage when mapped to the reference genome)?

9. Is much known about the background diversity of the ST17 strain that was causing this infection? Could this case have represented a wipeout of colonising bacteria with antibiotic treatment initially and re-invasion of the host respiratory space by different clones of the ST17 outbreak strain that were already present in the ITU environment, rather than selection/emergence of low-level mutants (similar to the secondary colonisation event proposed for the gut [lines 217-219])?

11. Figure 2D. The Muller plot is a really nice way of representing these data, but why does this only include a subset of all the mutants identified? e.g. why are the *bifA/soxA/oprD* mutant and *oprM* mutants (1/10 [10%] of isolates respectively at D21) not represented? Please represent the proportions of all the mutant populations at each timepoint. It would be helpful to have a similar plot for the GI tract isolates, which will make clear that the impact of antimicrobial administration on these two different anatomical compartments appears very different and is inconsistent.

Small grammatical issues picked up:

Lines 175-176 - suggest change "carried" to: "...the loss of the plasmid carrying the *aac4*..."

Lines 608-609 - suggest change "phenotypic" to: "...all data on isolate phenotypes will be"

Responses to reviewers comments of NCOMMS -20-38292.

Our responses are shown below in italics and sections of the manuscript that have either been added or significantly reworked are shown highlighted in the revised text.

Reviewer #1 (Remarks to the Author):

To understand population responses to antibiotic therapy, the authors sampled *P. aeruginosa* isolates (>100 from the infected respiratory site and from the gut) from one patient undergoing antibiotic therapy over 27 days and used WGS to characterize genetic changes that were associated with a raise of the bacterial population in the respiratory tract after discontinuation of antibiotic therapy. The authors convincingly demonstrate that the raise of the population following meropenem therapy was associated with the development of resistance (via the acquisition of mutations in *oprD* and *wbpM*). However the *oprD* mutants, which exhibited elevated resistance levels, were later replaced by mutants that showed inactivated multi-drug efflux pumps (anti-resistance phenotype). Their raise was probably due to their increased fitness as compared to the ancestral strain, which was constitutively overexpressing the efflux pump. The other (weaker) resistance determinant (mutation in *wbpM*), was not replaced. However, this mutant exhibited a fitness advantage over the ancestral strain.

The study is well-designed and provides a high-resolution characterization of bacterial responses to antibiotic treatment. It shows that antibiotic therapy selects for resistance conferring mutations and once discontinued, mutations that increase fitness predominate. Interestingly, these changes in the population was exclusively seen at the infected site (and not in the colonized gut).

1.1 I am not absolutely convinced that the results of this study “suggest that selection for anti-resistance mutations may be a common feature of *P. aeruginosa* infections” (line 275). There seems to be only a selection for anti-resistance, if the resistance conferring mutation was associated with a fitness burden – as in the case of *mexAB* pump overexpression. No selection for anti-resistance was observed for the *wbpM* mutation.

We agree with the reviewer that this is a misleading generalization of our results. In the revised manuscript, we have clarified this point by saying that selection for MexAB-OprM mutations appears to be widespread in Pseudomonas infections, and we have clearly stated that this is an interesting counter-example to established examples of compensatory evolution (line 256-262).

Specific comments:

1.2 Can the finding of a lack of an evolutionary response in the gut be correlated to the different population size in the gut? And can this be experientially shown by e.g.

treating large and small bottleneck populations of the ancestral isolate with meropenem in vitro?

Unfortunately it is very difficult to estimate total populations sizes of Pseudomonas in either the gut or lung by patient sampling. Polymorphism data could theoretically be used to estimate relative population size, but this would require us to assume that the respective populations were at mutation/drift equilibrium, which is very unlikely to be the case in this patient as this represents a new infection. As such, it would be very difficult to conclusively connect the results of the experiment proposed by the reviewer with the clinical data from the patient. However, we appreciate that understanding evolutionary responses to antibiotics in the gut and lung is an important issue. First, we now present data on the abundance of Pseudomonas in the gut over time (Figure 1B, line 120-124). Interestingly, the abundance of Pseudomonas did not decline during treatment, suggesting that meropenem was not effective in the gut. In support of this argument, we present evidence from the literature showing that meropenem concentrations are low in the gut relative to the lung (and we present experimental data showing that the meropenem MIC of the ancestral strain increases 4 fold under anaerobic culture conditions (line 153-169). Put simply, numerous lines of evidence suggest that meropenem was simply not effective against intestinal Pseudomonas, providing a simple explanation for the lack of an evolutionary response to meropenem treatment. We discuss the potential role of differential effects of antibiotics at different anatomical sites in the revised conclusion (line 409-423).

1.3 Did the authors find other gain-of fitness mutations (in addition to the efflux pump mutations) that became apparent in the late isolates?

No, we did not any other mutations in the late isolates other than a few singleton mutations that are unlikely to have been important (Figure 2B)

1.4 The greatest fitness increase should be observed in a mexAB-OprM / wbpM double mutant (pro and anti-resistant). Did the authors observe co-occurrence of the two mutations in single clinical isolates and what is the effect on fitness?

This is an interesting idea. We did not observe the co-occurrence of these mutations in a single isolate (Figure 2B). Measuring the fitness of this potential mutant would require us to reconstruct these mutations individually and in combination in the ancestral strain. We have not attempted this experiment, as the work required to carry out this genetic manipulation in a clinical isolate would be very substantial, and it is unclear how this fitness data would make an important contribution to our paper, especially given that these mutants were never observed.

Reviewer #2 (Remarks to the Author):

Thank you for asking me to review this interesting study by Wheatley et al, in which the authors characterise the population and genetic changes in cultured *Pseudomonas aeruginosa* isolates with a focus on mechanisms of antimicrobial resistance within a single patient admitted to critical care with a ventilator-associated pneumonia (VAP). In total they analysed n=107 *P. aeruginosa* isolates (n=59 from endotracheal aspirate samples; n=48 from peri-anal swabs) at 7 timepoints over a 21-day timeframe, using a combination of whole genome sequencing (WGS), growth rate assays to assess fitness, and antimicrobial susceptibility testing to assess resistance phenotypes.

They conclude: (i) bacterial populations in acute *P. aeruginosa* infections in the lung fluctuate rapidly in response to antimicrobial selection pressures; (ii) a short-course (2 days) of meropenem treatment drove the selection of deltaoprD (W829*) and deltawbpM (V1201G) mutants captured 8 days later (at D13); and (iii) subsequent replacement of oprD mutants with MexAB-OprM efflux pump mutants with relatively higher growth rates and decreased colistin resistance then occurred, once a 13-day treatment course of colistin had been completed at D14.

Overall, developing a better understanding of the real-life selection and persistence of resistance in bacterial populations is of great scientific and clinical interest, the design of the study is elegant, and the manuscript is well-written. However, whilst I appreciate that such detailed analyses of genotypic and phenotypic diversity are difficult and resource-intensive, caution is required in drawing generalizable conclusions from a study of these changes in a single individual (i.e. this is effectively a study of n=1), particularly when ancestral populations of the same clone in the GI tract of the same patient remained largely unaffected.

A consideration of other potentially relevant factors should perhaps be mentioned - for example:

2.1 • Antibiotic distributions and contributions in the site of infection being cultured (e.g. thought to be low for lung/epithelial lining fluid when IV colistin is administered <https://www.ncbi.nlm.nih.gov/pmc/articles/PMC3224467/>), which mean that it might be hard to interpret the likely in vivo impact of antimicrobials on these strains in the context of only in vitro MICs?

We thank the reviewer for this helpful suggestion. In the revised manuscript we highlight the complex pharmacokinetics of colistin and cite this reference (line 136-140;line 164-169). We also present evidence from the literature showing that concentrations of meropenem are high in the lung relative to the gut, and we show that anaerobic growth has a big impact on meropenem resistance (line 155-164). These pharmacokinetic and pharmacodynamic

considerations help to explain the lack of clinical (Figure 1B) or bacterial (Figure 1 C-E) response to meropenem therapy in the gut population.

2.2 How sample/sampling variability might impact on the accuracy of bacterial titre measurements?

This is an interesting point. The lung and gut samples were collected from the same anatomical site using the same method at different time points (line 453-478). However, without collecting multiple samples per site it is not possible to formally estimate the confidence in titre measurements. Our analysis of this data places emphasis on large, overall trends in for changes in bacterial titre (ie >10x changes in density) and population composition (see 2.3 below). Also, the correlation between population density (Figure 1 A,B) or composition (Figure 3A) are very similar, as we would expect if these trends captured underlying dynamics of the bacterial population as opposed to sampling error.

2.3 How characterising the diversity amongst small numbers of isolates when compared with the vast number of isolates present (~9-12 isolates at each sampling event) can then reliably be used to extrapolate to determining the distributions of these mutants at the bacterial population-level. My understanding is that we cannot be sure how many colonies you need to sequence to accurately reflect the distribution of various strains in a sample, but it would probably be more than 12, especially in the GI tract?

The number of colonies that you need to sequence to accurately measure the prevalence of a particular strain depends on the prevalence of the strain (confidence in proportion data increases as proportions approach 0.5) and there is no a priori reason why you should need to sequence more colonies from one anatomical site than another. However, we agree with reviewer that this is an important point to address, as our initial submission did not include estimates of the uncertainty in our estimated mutant frequencies. In the revised manuscript we have highlighted the limited number of colonies sequenced per time point (line 215-217), and we have shown mutant frequencies along with 90% confidence intervals (Figure 3A). Given the uncertainty in estimates of mutant frequency per time point we have focused our interpretation of this data on 2 clear overall trends in the mutant frequency data (line 212-268). At the same time, it is worth emphasizing that microbiology labs generally only analyze 1 isolate per patient sample, and our analysis of >10; total >100 isolates allowed us to capture diversity and dynamics that would not have been possible to capture using single isolate sampling.

2.4 What role the human immune response might have in the selection/adaptation of various sub-strains?

This is a very important point, and we added substantial data to the manuscript to address this point. To measure the host response to infection,

we quantified levels of inflammatory cytokines in samples of endotracheal during this patient's stay in ICU (Figure 5 A-E , lines 315-329). Importantly, all of these cytokines have previously been shown to provide protection against P.aeruginosa lung infection. This data shows that both the first and second waves of infection in the patient were associated with elevated expression of inflammatory biomarkers, including neutrophil attracting factor (IL-8). To understand the functional consequence of host immunity, we measured the impact of two key host antimicrobial peptides (HDB-3 and LL37) that are upregulated as part of the inflammatory response on bacterial isolates (Figure 5F). We found that P.aeruginosa isolates were sensitive to LL37, but not HDB-3. Interestingly, we found a small, but statistically significant increase in LL37 tolerance in the mutants that were selected in vivo compared to the ancestral strain, suggesting that increased resistance to host immunity may have contributed to the benefit associated with these mutations during infection (line 331-346).

2.5 Whether differential gene expression might have a role to play?

This point is unclear. It is possible that differential expression of bacterial genes played a role in mediating responses to antibiotic treatment, but I would be a bit skeptical about trying to assess differences in gene expression by doing RNAseq on isolates bacterial colonies that were cultured in vitro. However, we mention the potential role of differential gene expression in mediating high levels of in vivo resistance to colistin (line 139-140).

2.6 What interaction might be occurring between the Pseudomonas sub-strains and other bacteria (both cultured and uncultured) that might affect selection? The culture results in Fig.1A, suggest that beyond D13 other non-Pseudomonas culturable bacteria are clearly present?

At the time of antibiotic treatment, Pseudomonas aeruginosa was the only culturable bacterium that was detected in the lung microbiome. The reviewer is correct in pointing out that the second wave of Pseudomonas infection was associated with establishment of a culturable lung microbiome, and we have provided rank order abundance data on changes in the culturable microbiome over time (Figure 1A, lines 111-114). However, we have not carried out any functional assays on the interactions between Pseudomonas and the culturable lung microbiome, as isolates of the lung microbiome were not kept. It is conceivable that interactions between Pseudomonas and the lung microbiome helped to shape the evolutionary responses that were observed during the second wave of the infection, and we have mentioned this limitation in the revised manuscript (line 403-408). However, all of the responses that we observed during this second wave can be explained by the selection for high growth rate (which led to the fall of oprD), colistin tolerance (which may have delayed the emergence of MexAB-OprM mutants). Given

this, we argue that the quantifying Pseudomonas-microbiome interactions is not necessary to understand the responses to antibiotic treatment.

2.7 Metagenomics might be a complementary, additional way to look more broadly for strain-level diversity that is not identified by picking a small subset of colonies for genomic evaluation (e.g. by using strain-aware metagenomic profilers, such as <https://pubmed.ncbi.nlm.nih.gov/28167665/>). This could be done either on sweeps of cultured selected growth, and/or on unselected growth to evaluate wider bacterial species diversity in samples. Transcriptomics might be a useful way to understand genome-wide differences in gene expression that may be impacting on selection/adaptation that are not reflected in DNA sequencing analyses.

Pseudomonas aeruginosa: The reviewer is certainly correct in asserting that metagenomic approaches could have potentially allowed us to better quantify the frequency of SNPs in the Pseudomonas aeruginosa population and to identify rare variants. We chose to focus on isolate sequencing. First, this allowed us to identify linkage between SNPs, which allowed us to confidently reconstruct the phylogeny of our isolates, which is key to the interpretation of our data (Figure 2B, line 196-211). Second, isolate sequencing also allowed us to link isolate phenotypes and genotypes, which is also key to our understanding of resistance in this system (Figure 3B, 3C; line 212-267). The ideal experiment in a world unconstrained by time and research budgets would be to combine long-read metagenomic sequencing with sequencing of individual isolates, but this was simply not possible in this case. Our isolate data suggests that metagenomic approaches may not have yielded many new insights into Pseudomonas, as the evolutionary dynamics were mainly driven by a small number (3) of common variants that were present at high frequencies (>10%) for most of the infection.

Microbiome: We agree with the reviewer that metagenomic approaches would have produced a much more refined view of the dynamics of the lung microbiome than we obtained from measuring changes in the rank-abundance of culturable bacteria (Figure 1A). However, it not obvious how this data would have helped us to understand the responses of P.aeruginosa to antibiotic treatment. We argue that all of the main responses of the Pseudomonas population during this infection can be understood by the effects of antibiotics, selection for high growth rate and/or resistance, and host immunity without taking the lung microbiome into consideration. The microbiome field is full of studies that have quantified correlations between microbiome composition and antibiotic treatment, but the mechanisms underpinning these correlations are usually poorly understood. The great advantage of our approach is that we have been able to use experiments to test the drivers of resistance, and it is unclear how we could actually test the importance of an altered lung

microbiome on antibiotic resistance in Pseudomonas aeruginosa (particularly in the case of unculturable bacteria!)

Transcriptomics: see response to point 2.5 above.

In addition to the above, please consider the following comments:

(2.8) 1. I think the plan was to analyse 12 isolates per sample, and nine samples were evaluated (n=5 ETA, 4 peri-rectal) - why then are there only 107/108 isolates sequenced (seems to affect lung sample at D21)? Was there a sequencing failure? Please explain and include a short statement as to why this isolate was excluded.

One of the isolates turned out to be S.epidermidis, and we excluded this isolate from all analyses (line 474-476).

(2.9) 2. Ancestral P. aeruginosa populations in the lung appear to be declining before piperacillin-tazobactam and colistin treatment starts (decrease in counts from 1×10^6 to 1×10^4 CFU/mL - Fig. 1A from days 1>2) - what do the authors propose is the cause of this?

This is an excellent point, and we have addressed this in the revised manuscript (line 316-320). The titre of Pseudomonas declined by >10 fold prior to any antibiotic treatment and in the absence of any competing microbes. Crucially, we show that the host inflammatory response had already been induced on day 1 (Figure 5 B-E) and we show that the ancestral strain was sensitive to LL-37, a key effector of inflammatory response (Figure 5F).

(2.10) 3. Why is there a delay in the emergence of meropenem-R mutants if meropenem is thought to be the main selection pressure driving this emergence (meropenem administration stops on D4, mutants appear to emerge after D8),

This is a very important point that we did not fully address in the initial submission. We have used root-to-tip regression to estimate the time to the most recent common ancestor (MRCA) of all of the isolates that were detected (Figure 2C; line 195-211). We estimate that the MRCA was present at day 0, providing good evidence that all of the variants that we detected evolved in situ (with the exception of two lung isolates that differ from the MRCA by a large number of SNPs). Given this, we argue that the difference in time between meropenem treatment and the time of detection of meropenem resistant mutants reflects the time needed for single cells of meropenem resistant mutants to expand to form detectable populations (minimum detectable density $>10^2$ CFU/mL). This point is addressed in the revised manuscript (line 219-222) and we argue that the host immune response is likely to have contributed to the lag between meropenem treatment and the detection of resistant mutants (line 367-379).

(2.11)...and why should meropenem represent a significant selection pressure for these mutants in the first place given that the ancestral isolates already have resistant MICs at baseline (>8 mg/L, Fig 1C)? I can't find any evidence of a post-antibiotic effect for meropenem that lasts beyond several hours.

This is a very good point, and we have addressed this issue in the revised manuscript (line 143-152). In this case, the patient was actually treated with both meropenem and colistin (Figure 1A), and previous work has shown that these antibiotics have synergistic effects in Pseudomonas aeruginosa. We tested the impact of colistin/meropenem combination therapy using a checkerboard assay. Importantly, we found that sub-MIC doses of colistin reduced the concentration of meropenem needed to inhibit growth by a factor of 4. In other words, our data suggests that colistin potentiated the effect of meropenem, generating selection for resistance.

(2.12) 4. Given wbpM's proposed role in LPS synthesis (as in ref 26 in the manuscript), and colistin's mode of action (interaction with LPS and then destabilisation of the cell membrane), isn't sub-therapeutic colistin as a selection pressure driving the emergence of the wbpM mutants a possible alternative explanation?

Although the role of wbpM mutations suggests that they have provided resistance to colistin, we found no increase in MIC or colistin tolerance associated with the wbpM mutation (Figure 1E,F; Figure 3C).

(2.13) Colistin susceptibility testing remains a slightly unreliable business, we are not clear on the in vivo drug concentrations achieved in the lung (e.g. <https://academic.oup.com/cid/advance-article/doi/10.1093/cid/ciaa121/5735218>), and these mutant populations emerged during colistin (not meropenem) monotherapy. Notably, the mexAB-oprM mutants (significantly less colistin-tolerant) only emerge after colistin therapy has stopped, and at this point briefly out-compete the wbpM mutant, probably because of faster growth rates.

We agree with the referee that our data suggests that continued colistin therapy delayed the emergence of MexAB-OprM mutants, and we have included this point in the revised manuscript along with the helpful citation suggested by the referee (Figure 3C; line 256-267).

(2.14) Despite the in vitro colistin testing findings, is this really population adaptation in response to meropenem and colistin therapy, and not just meropenem administration?

This is another very good point that we have addressed in a number of ways. As outlined in 2.11 we have shown that the combination of meropenem and colistin was more effective at inhibiting the ancestral strain than meropenem alone, and we have also highlighted the role of colistin in delaying the emergence of MexAB-OprM mutants (2.13)

(2.15) 5. The authors mention this briefly but don't hypothesise why/answer the question in the text - if antibiotic selection pressures are the critical driving force, why was the same clonal ancestral gastrointestinal population of *P. aeruginosa* in this patient unaffected by the same antibiotic exposures?

We have addressed this important point (see response to 1.2)

(2.16) 6. Lines 171-176 - With respect to the accessory genome analysis (based here on using Prokka annotations and GenAPI to identify presence/absence of annotated genes) - would it be worthwhile undertaking a k-mer based evaluation on non-core reads to determine if there is any variability that is not captured by looking only at annotated regions (e.g. in promoters etc.)? Previous analyses e.g. <https://academic.oup.com/gbe/article/11/5/1385/5454722> suggest that mutations in intergenic regions (i.e. potentially more likely to be un-annotated) might be very important in adaptation and "are directly responsible for the evolution of important pathogenic phenotypes including antibiotic sensitivity".

Our analysis was based on the entire genome, including both coding and non-coding regions, and we detected a small number of intergenic SNPs, none of which were associated with elevated antibiotic resistance. We have clarified this important point in the revised manuscript. We also analyzed changes in genome composition, and the only difference that we found was the absence of a single gene in a single intestinal isolate (line 185-193).

(2.17)7. Lines 189-192 - the authors describe 12 isolates that were cultured on D13 from the ETA samples. Please describe/mention the other genotype (indel in oprD from what I can see in Fig.2B)?

We have removed this section of the text to put a greater focus on overall patterns of change in genotype frequency. The remaining 6 isolates from this time point carried the wbpM mutation.

(2.18)8. Was there any evidence that gene duplication might be playing a role in adaptation (could perhaps be evaluated from looking at read coverage when mapped to the reference genome)?

We did not detect any evidence of duplication of chromosomal genes. However, we found evidence of subtle, but pervasive, variation in the copy number of the plasmid that was carried by all isolates (Figure 4C). To understand the functional importance of this variation, we transformed the plasmid into the PA01 model strain and measured the impact of plasmid carriage on antibiotic resistance and growth rate. We found that the plasmid carriage conferred large increases in resistance to meropenem and piperacillin tazobactam (Figure 4A), but plasmid carriage came also generate a fitness cost (Figure 4C). This trade-off between resistance and fitness explains why

plasmid copy number declined over time during the course of infection. This decline in plasmid copy number was not associated with any plasmid or chromosomal mutations, suggesting that altered copy number was driven by inherent variation in plasmid copy number (Figure 2B). Variation in plasmid copy number is likely to be ubiquitous and this represents (to the best of our knowledge) the first evidence for selection on plasmid copy number variation during an infection. We present these results in the main text (line 270-313) and in the conclusion (line 384-388) of the revised manuscript.

(2.19) 9. Is much known about the background diversity of the ST17 strain that was causing this infection? Could this case have represented a wipeout of colonising bacteria with antibiotic treatment initially and re-invasion of the host respiratory space by different clones of the ST17 outbreak strain that were already present in the ITU environment, rather than selection/emergence of low-level mutants (similar to the secondary colonisation event proposed for the gut [lines 217-219])?

This infection occurred in the context of an outbreak of ST17 in the ICU of the Virgen Macarena hospital (line 183-183). We have addressed this important question of in situ evolution vs. re-infection in our response to point 2.7. To briefly re-cap, we estimate that all of the isolates (with the exception of the two outliers mentioned above) were descended from a single clonal ancestor that initiated the infection at day 0, strongly supporting the idea that diversity in the patient reflects in situ evolution rather than continual re-infection.

(2.20) 11. Figure 2D. The Muller plot is a really nice way of representing these data, but why does this only include a subset of all the mutants identified? e.g. why are the *bifA/soxA/oprD* mutant and *oprM* mutants (1/10 [10%] of isolates respectively at D21) not represented? Please represent the proportions of all the mutant populations at each timepoint.

Our initial Muller plot included only SNPs present in >1 isolate, and we have now revised this figure to include the singleton SNPs mentioned above (Figure 6).

(2.21) It would be helpful to have a similar plot for the GI tract isolates, which will make clear that the impact of antimicrobial administration on these two different anatomical compartments appears very different and is inconsistent.

*In the revised manuscript we have included a plot showing estimates of the abundance of *P.aeruginosa* and total culturable bacteria in the gut (Figure 1B). This plot clearly shows that antibiotic treatment did not suppress the intestinal population of *P.aeruginosa*, and we highlight this point in the revised manuscript (line 120-123; line 153-169). We also discuss the potential importance of differential effects of antibiotic treatment on the transmission and infection biology of *P.aeruginosa* (line 409-423).*

Small grammatical issues picked up:

Lines 175-176 - suggest change "carried" to: "...the loss of the plasmid carrying the aac4..."

Lines 608-609 - suggest change "phenotypic" to: "...all data on isolate phenotypes will be"

REVIEWERS' COMMENTS

Reviewer #1 (Remarks to the Author):

The authors adequately addressed all comments/questions.

Reviewer #2 (Remarks to the Author):

Many thanks to the authors for addressing my thoughts and comments. I have no further queries. This is a very interesting and thorough piece of work and it has been a pleasure to review it.